# Improving robustness to corruptions with multiplicative weight perturbations

**Trung Trinh**[1]     **Markus Heinonen**[1]     **Luigi Acerbi**[2]     **Samuel Kaski**[1,3]

[1]Department of Computer Science, Aalto University, Finland
[2]Department of Computer Science, University of Helsinki, Finland
[3]Department of Computer Science, University of Manchester, United Kingdom
`{trung.trinh, markus.o.heinonen, samuel.kaski}@aalto.fi,`
`luigi.acerbi@helsinki.fi`

## Abstract

Deep neural networks (DNNs) excel on clean images but struggle with corrupted ones. Incorporating specific corruptions into the data augmentation pipeline can improve robustness to those corruptions but may harm performance on clean images and other types of distortion. In this paper, we introduce an alternative approach that improves the robustness of DNNs to a wide range of corruptions without compromising accuracy on clean images. We first demonstrate that input perturbations can be mimicked by multiplicative perturbations in the weight space. Leveraging this, we propose Data Augmentation via Multiplicative Perturbation (DAMP), a training method that optimizes DNNs under random multiplicative weight perturbations. We also examine the recently proposed Adaptive Sharpness-Aware Minimization (ASAM) and show that it optimizes DNNs under adversarial multiplicative weight perturbations. Experiments on image classification datasets (CIFAR-10/100, Tiny-ImageNet and ImageNet) and neural network architectures (ResNet50, ViT-S/16, ViT-B/16) show that DAMP enhances model generalization performance in the presence of corruptions across different settings. Notably, DAMP is able to train a ViT-S/16 on ImageNet from scratch, reaching the top-1 error of $23.7\%$ which is comparable to ResNet50 without extensive data augmentations.[1]

## 1   Introduction

Deep neural networks (DNNs) demonstrate impressive accuracy in computer vision tasks when evaluated on carefully curated and clean datasets. However, their performance significantly declines when test images are affected by natural distortions such as camera noise, changes in lighting and weather conditions, or image compression algorithms (Hendrycks and Dietterich, 2019). This drop in performance is problematic in production settings, where models inevitably encounter such perturbed inputs. Therefore, it is crucial to develop methods that produce reliable DNNs robust to common image corruptions, particularly for deployment in safety-critical systems (Amodei et al., 2016).

To enhance robustness against a specific corruption, one could simply include it in the data augmentation pipeline during training. However, this approach can diminish performance on clean images and reduce robustness to other types of corruptions (Geirhos et al., 2018). More advanced data augmentation techniques (Cubuk et al., 2018; Hendrycks et al., 2019; Lopes et al., 2019) have been developed which effectively enhance corruption robustness without compromising accuracy on clean images. Nonetheless, a recent study by Mintun et al. (2021) has identified a new set of image corruptions to which models trained with these techniques remain vulnerable. Besides data

---

[1]Our code is available at `https://github.com/trungtrinh44/DAMP`

38th Conference on Neural Information Processing Systems (NeurIPS 2024).

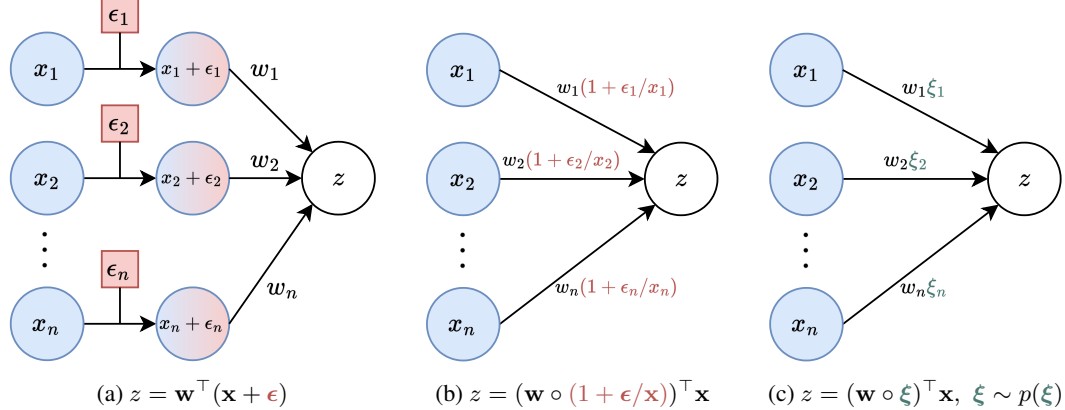

(a) $z = \mathbf{w}^\top(\mathbf{x} + \boldsymbol{\epsilon})$     (b) $z = (\mathbf{w} \circ (1 + \boldsymbol{\epsilon}/\mathbf{x}))^\top \mathbf{x}$     (c) $z = (\mathbf{w} \circ \boldsymbol{\xi})^\top \mathbf{x}, \; \boldsymbol{\xi} \sim p(\boldsymbol{\xi})$

Figure 1: **Depictions of a pre-activation neuron $z = \mathbf{w}^\top \mathbf{x}$ in the presence of (a) covariate shift $\boldsymbol{\epsilon}$, (b) a multiplicative weight perturbation (MWP) equivalent to $\boldsymbol{\epsilon}$, and (c) random MWPs $\boldsymbol{\xi}$.** $\circ$ denotes the Hadamard product. Figs. (a) and (b) show that for a covariate shift $\boldsymbol{\epsilon}$, one can always find an equivalent MWP. From this intuition, we propose to inject random MWPs $\boldsymbol{\xi}$ to the forward pass during training as shown in Fig. (c) to robustify a DNN to covariate shift.

augmentation, ensemble methods such as Deep ensembles and Bayesian neural networks have also been shown to improve generalization in the presence of corruptions (Lakshminarayanan et al., 2017; Ovadia et al., 2019; Dusenberry et al., 2020; Trinh et al., 2022). However, the training and inference costs of these methods increase linearly with the number of ensemble members, rendering them less suitable for very large DNNs.

**Contributions** In this work, we show that simply perturbing weights with multiplicative random variables during training can significantly improve robustness to a wide range of corruptions. Our contributions are as follows:

- We show in Section 2 and Fig. 1 that the effects of input corruptions can be simulated during training via multiplicative weight perturbations.

- From this insight, we propose a new training algorithm called Data Augmentation via Multiplicative Perturbations (DAMP) which perturbs weights using multiplicative Gaussian random variables during training while having the same training cost as standard SGD.

- In Section 3, we show a connection between adversarial multiplicative weight perturbations and Adaptive Sharpness-Aware Minimization (ASAM) (Kwon et al., 2021).

- Through a rigorous empirical study in Section 4, we demonstrate that DAMP consistently improves generalization ability of DNNs under corruptions across different image classification datasets and model architectures.

- Notably, we demonstrate that DAMP can train a Vision Transformer (ViT) (Dosovitskiy et al., 2021) from scratch on ImageNet, achieving similar accuracy to a ResNet50 (He et al., 2016a) in 200 epochs with only basic Inception-style preprocessing (Szegedy et al., 2016). This is significant as ViT typically requires advanced training methods or sophisticated data augmentation to match ResNet50's performance when being trained on ImageNet from scratch (Chen et al., 2022; Beyer et al., 2022). We also show that DAMP can be combined with modern augmentation techniques such as MixUp (Zhang et al., 2018) and RandAugment (Cubuk et al., 2020) to further improve robustness of neural networks.

## 2   Data Augmentation via Multiplicative Perturbations

In this section, we demonstrate the equivalence between input corruptions and multiplicative weight perturbations (MWPs), as shown in Fig. 1, motivating the use of MWPs for data augmentation.

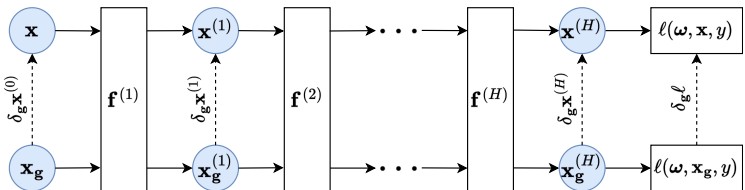

Figure 2: **Depiction of how a corruption g affects the output of a DNN.** Here $\mathbf{x_g} = \mathbf{g}(\mathbf{x})$. The corruption $\mathbf{g}$ creates a shift $\boldsymbol{\delta_g}\mathbf{x}^{(0)} = \mathbf{x_g} - \mathbf{x}$ in the input $\mathbf{x}$, which propagates into shifts $\boldsymbol{\delta_g}\mathbf{x}^{(h)}$ in the output of each layer. This will eventually cause a shift in the loss $\boldsymbol{\delta_g}\ell$. This figure explains why the model performance tends to degrade under corruption.

## 2.1 Problem setting

Given a training data set $\mathcal{S} = \{(\mathbf{x}_k, y_k)\}_{k=1}^{N} \subseteq \mathcal{X} \times \mathcal{Y}$ drawn i.i.d. from the data distribution $\mathcal{D}$, we seek to learn a model that generalizes well on both clean and corrupted inputs. We denote $\mathcal{G}$ as a set of functions whose each member $\mathbf{g} : \mathcal{X} \to \mathcal{X}$ represents an input corruption. That is, for each $\mathbf{x} \in \mathcal{X}$, $\mathbf{g}(\mathbf{x})$ is a corrupted version of $\mathbf{x}$.[2] We define $\mathbf{g}(\mathcal{S}) := \{(\mathbf{g}(\mathbf{x}_k), y_k)\}_{k=1}^{N}$ as the training set corrupted by $\mathbf{g}$. We consider a DNN $\mathbf{f} : \mathcal{X} \to \mathcal{Y}$ parameterized by $\boldsymbol{\omega} \in \mathcal{W}$. Given a per-sample loss $\ell : \mathcal{W} \times \mathcal{X} \times \mathcal{Y} \to \mathbb{R}_+$, the training loss is defined as the average loss over the samples $\mathcal{L}(\boldsymbol{\omega}; \mathcal{S}) := \frac{1}{N}\sum_{k=1}^{N} \ell(\boldsymbol{\omega}, \mathbf{x}_k, y_k)$. Our goal is to find $\boldsymbol{\omega}$ which minimizes:

$$\mathcal{L}(\boldsymbol{\omega}; \mathcal{G}(\mathcal{S})) := \mathbb{E}_{\mathbf{g}\sim\mathcal{G}}[\mathcal{L}(\boldsymbol{\omega}; \mathbf{g}(\mathcal{S}))] \tag{1}$$

without knowing exactly the types of corruption contained in $\mathcal{G}$. This problem is crucial for the reliable deployment of DNNs, especially in safety-critical systems, since it is difficult to anticipate all potential types of corruption the model might encounter in production.

## 2.2 Multiplicative weight perturbations simulate input corruptions

To address the problem above, we make two key assumptions about the corruptions in $\mathcal{G}$:

**Assumption 1** (Bounded corruption). *For each corruption function* $\mathbf{g} : \mathcal{X} \to \mathcal{X}$ *in* $\mathcal{G}$, *there exists a constant* $M > 0$ *such that* $\|\mathbf{g}(\mathbf{x}) - \mathbf{x}\|_2 \leq M$ *for all* $\mathbf{x} \in \mathcal{X}$.

**Assumption 2** (Transferable robustness). *A model's robustness to corruptions in* $\mathcal{G}$ *can be indirectly enhanced by improving its resilience to a more easily simulated set of input perturbations.*

Assumption 1 implies that the corrupted versions of an input $\mathbf{x}$ must be constrained within a bounded neighborhood of $\mathbf{x}$ in the input space. Assumption 2 is corroborated by Rusak et al. (2020), who demonstrated that distorting training images with Gaussian noise improves a DNN's performance against various types of corruption. We further validate this observation for corruptions beyond Gaussian noise in Section 4.1. However, Section 4.1 also reveals that using corruptions as data augmentation degrades model performance on clean images. Consequently, we need to identify a method that efficiently simulates diverse input corruptions during training, thereby robustifying a DNN against a wide range of corruptions without compromising its performance on clean inputs.

One such method involves injecting random multiplicative weight perturbations (MWPs) into the forward pass of DNNs during training. The intuition behind this approach is illustrated in Fig. 1. Essentially, for a pre-activated neuron $z = \mathbf{w}^\top \mathbf{x}$ in a DNN, given a corruption causing a covariate shift $\boldsymbol{\epsilon}$ in the input $\mathbf{x}$, Figs. 1a and 1b show that one can always find an equivalent MWP $\boldsymbol{\xi}(\boldsymbol{\epsilon}, \mathbf{x})$:

$$z = \mathbf{w}^\top(\mathbf{x} + \boldsymbol{\epsilon}) = (\mathbf{w} \circ \boldsymbol{\xi}(\boldsymbol{\epsilon}, \mathbf{x}))^\top \mathbf{x}, \quad \boldsymbol{\xi}(\boldsymbol{\epsilon}, \mathbf{x}) = 1 + \boldsymbol{\epsilon}/\mathbf{x} \tag{2}$$

where $\circ$ denotes the Hadamard product. This observation suggests that input corruptions can be simulated during training by injecting random MWPs into the forward pass, as depicted in Fig. 1c, resulting in a model more robust to corruption. We thus move the problem of simulating corruptions from the input space to the weight space.

Here we provide theoretical arguments supporting the usage of MWPs to robustify DNNs. To this end, we study how corruption affects training loss. We consider a feedforward neural network $\mathbf{f}(\mathbf{x}; \boldsymbol{\omega})$

---

[2]For instance, if $\mathbf{x}$ is a clean image then $\mathbf{g}(\mathbf{x})$ could be $\mathbf{x}$ corrupted by *Gaussian noise*.

of depth $H$ parameterized by $\boldsymbol{\omega} = \{\mathbf{W}^{(h)}\}_{h=1}^{H} \in \mathcal{W}$, which we define recursively as follows:

$$\mathbf{f}^{(0)}(\mathbf{x}) := \mathbf{x}, \quad \mathbf{z}^{(h)}(\mathbf{x}) := \mathbf{W}^{(h)}\mathbf{f}^{(h-1)}(\mathbf{x}), \quad \mathbf{f}^{(h)}(\mathbf{x}) := \boldsymbol{\sigma}^{(h)}(\mathbf{z}^{(h)}(\mathbf{x})), \quad \forall h = 1, \ldots, H \quad (3)$$

where $\mathbf{f}(\mathbf{x}; \boldsymbol{\omega}) := \mathbf{f}^{(H)}(\mathbf{x})$ and $\boldsymbol{\sigma}^{(h)}$ is the non-linear activation of layer $h$. For brevity, we use $\mathbf{x}^{(h)}$ and $\mathbf{x}_{\mathbf{g}}^{(h)}$ as shorthand notations for $\mathbf{f}^{(h)}(\mathbf{x})$ and $\mathbf{f}^{(h)}(\mathbf{g}(\mathbf{x}))$ respectively. Given a corruption function $\mathbf{g}$, Fig. 2 shows that $\mathbf{g}$ creates a covariate shift $\boldsymbol{\delta}_{\mathbf{g}}\mathbf{x}^{(0)} := \mathbf{x}_{\mathbf{g}}^{(0)} - \mathbf{x}^{(0)}$ in the input $\mathbf{x}$ leading to shifts $\boldsymbol{\delta}_{\mathbf{g}}\mathbf{x}^{(h)} := \mathbf{x}_{\mathbf{g}}^{(h)} - \mathbf{x}^{(h)}$ in the output of each layer. This will eventually cause a shift in the per-sample loss $\boldsymbol{\delta}_{\mathbf{g}}\ell(\boldsymbol{\omega}, \mathbf{x}, y) := \ell(\boldsymbol{\omega}, \mathbf{x}_{\mathbf{g}}, y) - \ell(\boldsymbol{\omega}, \mathbf{x}, y)$. The following lemma characterizes the connection between $\boldsymbol{\delta}_{\mathbf{g}}\ell(\boldsymbol{\omega}, \mathbf{x}, y)$ and $\boldsymbol{\delta}_{\mathbf{g}}\mathbf{x}^{(h)}$:

**Lemma 1.** *For all $h = 1, \ldots, H$ and for all $\mathbf{x} \in \mathcal{X}$, there exists a scalar $C_{\mathbf{g}}^{(h)}(\mathbf{x}) > 0$ such that:*

$$\boldsymbol{\delta}_{\mathbf{g}}\ell(\boldsymbol{\omega}, \mathbf{x}, y) \leq \left\langle \nabla_{\mathbf{z}^{(h+1)}}\ell(\boldsymbol{\omega}, \mathbf{x}, y) \otimes \boldsymbol{\delta}_{\mathbf{g}}\mathbf{x}^{(h)}, \mathbf{W}^{(h+1)} \right\rangle_F + \frac{C_{\mathbf{g}}^{(h)}(\mathbf{x})}{2}\|\mathbf{W}^{(h)}\|_F^2 \quad (4)$$

Here $\otimes$ denotes the outer product of two vectors, $\langle \cdot, \cdot \rangle_F$ denotes the Frobenius inner product of two matrices of the same dimension, $\| \cdot \|_F$ is the Frobenius norm, and $\nabla_{\mathbf{z}^{(h)}}\ell(\boldsymbol{\omega}, \mathbf{x}, y)$ is the Jacobian of the per-sample loss with respect to the pre-activation output $\mathbf{z}^{(h)}(\mathbf{x})$ at layer $h$. To prove Lemma 1, we use Assumption 1 and the following assumption about the loss function:

**Assumption 3** (Lipschitz-continuous objective input gradients). *The input gradient of the per-sample loss $\nabla_{\mathbf{x}}\ell(\boldsymbol{\omega}, \mathbf{x}, y)$ is Lipschitz continuous.*

Assumption 3 allows us to define a quadratic bound of the loss function using a second-order Taylor expansion. The proof of Lemma 1 is provided in Appendix A. Using Lemma 1, we prove Theorem 1, which bounds the training loss in the presence of corruptions using the training loss under multiplicative perturbations in the weight space:

**Theorem 1.** *For a function $\mathbf{g} : \mathcal{X} \rightarrow \mathcal{X}$ satisfying Assumption 1 and a loss function $\mathcal{L}$ satisfying Assumption 3, there exists $\boldsymbol{\xi}_{\mathbf{g}} \in \mathcal{W}$ and $C_{\mathbf{g}} > 0$ such that:*

$$\mathcal{L}(\boldsymbol{\omega}; \mathbf{g}(\mathcal{S})) \leq \mathcal{L}(\boldsymbol{\omega} \circ \boldsymbol{\xi}_{\mathbf{g}}; \mathcal{S}) + \frac{C_{\mathbf{g}}}{2}\|\boldsymbol{\omega}\|_F^2 \quad (5)$$

We provide the proof of Theorem 1 in Appendix B. This theorem establishes an upper bound for the target loss in Eq. (1):

$$\mathcal{L}(\boldsymbol{\omega}; \mathcal{G}(\mathcal{S})) \leq \mathbb{E}_{\mathbf{g} \sim \mathcal{G}}\left[ \mathcal{L}(\boldsymbol{\omega} \circ \boldsymbol{\xi}_{\mathbf{g}}; \mathcal{S}) + \frac{C_{\mathbf{g}}}{2}\|\boldsymbol{\omega}\|_F^2 \right] \quad (6)$$

This bound implies that training a DNN using the following loss function:

$$\mathcal{L}_{\boldsymbol{\Xi}}(\boldsymbol{\omega}; \mathcal{S}) := \mathbb{E}_{\boldsymbol{\xi} \sim \boldsymbol{\Xi}}\left[ \mathcal{L}(\boldsymbol{\omega} \circ \boldsymbol{\xi}; \mathcal{S}) \right] + \frac{\lambda}{2}\|\boldsymbol{\omega}\|_F^2 \quad (7)$$

where the expected loss is taken with respect to a distribution $\boldsymbol{\Xi}$ of random MWPs $\boldsymbol{\xi}$, will minimize the upper bound of the loss $\mathcal{L}(\boldsymbol{\omega}; \hat{\mathcal{G}}(\mathcal{S}))$ of a hypothetical set of corruptions $\hat{\mathcal{G}}$ simulated by $\boldsymbol{\xi} \sim \boldsymbol{\Xi}$. This approach results in a model robust to these simulated corruptions, which, according to Assumption 2, could indirectly improve robustness to corruptions in $\mathcal{G}$.

We note that the second term in Eq. (7) is the $L_2$-regularization commonly used in optimizing DNNs. Based on this proxy loss, we propose Algorithm 1 which minimizes the objective function in Eq. (7) when $\boldsymbol{\Xi}$ is an isotropic Gaussian distribution $\mathcal{N}(\mathbf{1}, \sigma^2\mathbf{I})$. We call this algorithm Data Augmentation via Multiplicative Perturbations (DAMP), as it uses random MWPs during training to simulate input corruptions, which can be viewed as data augmentations.

**Remark**  The standard method to calculate the expected loss in Eq. (7), which lacks a closed-form solution, is the Monte Carlo (MC) approximation. However, the training cost of this approach scales linearly with the number of MC samples. To match the training cost of standard SGD, Algorithm 1 divides each data batch into $M$ equal-sized sub-batches (Line 6) and calculates the loss on each sub-batch with different multiplicative noises from the noise distribution $\boldsymbol{\Xi}$ (Lines 7–9). The final gradient is obtained by averaging the sub-batch gradients (Line 11). Algorithm 1 is thus suitable for data parallelism in multi-GPU training, where the data batch is evenly distributed across $M > 1$ GPUs. Compared to SGD, Algorithm 1 requires only two additional operations: generating Gaussian samples and point-wise multiplication, both of which have negligible computational costs. In our experiments, we found that both SGD and DAMP had similar training times.

**Algorithm 1** DAMP: Data Augmentation via Multiplicative Perturbations
___
1: **Input:** training data $\mathcal{S} = \{(\mathbf{x}_k, y_k)\}_{k=1}^N$, a neural network $\mathbf{f}(\cdot; \boldsymbol{\omega})$ parameterized by $\boldsymbol{\omega} \in \mathbb{R}^P$, number of iterations $T$, step sizes $\{\eta_t\}_{t=1}^T$, number of sub-batch $M$, batch size $B$ divisible by $M$, a noise distribution $\boldsymbol{\Xi} = \mathcal{N}(\mathbf{1}, \sigma^2 \mathbf{I}_P)$, weight decay coefficient $\lambda$, a loss function $\mathcal{L}: \mathbb{R}^P \to \mathbb{R}_+$.
2: **Output:** Optimized parameter $\boldsymbol{\omega}^{(T)}$.
3: Initialize parameter $\boldsymbol{\omega}^{(0)}$.
4: **for** $t = 1$ **to** $T$ **do**
5:      Draw a mini-batch $\mathcal{B} = \{(\mathbf{x}_b, y_b)\}_{b=1}^B \sim \mathcal{S}$.
6:      Divide the mini-batch into $M$ *disjoint sub-batches* $\{\mathcal{B}_m\}_{m=1}^M$ of equal size.
7:      **for** $m = 1$ **to** $M$ **in parallel do**
8:          Draw a noise sample $\boldsymbol{\xi}_m \sim \boldsymbol{\Xi}$.
9:          Compute the gradient $\mathbf{g}_m = \nabla_{\boldsymbol{\omega}} \mathcal{L}(\boldsymbol{\omega}; \mathcal{B}_m)\big|_{\boldsymbol{\omega}^{(t)} \circ \boldsymbol{\xi}}$.
10:      **end for**
11:      Compute the average gradient: $\mathbf{g} = \frac{1}{M} \sum_{m=1}^M \mathbf{g}_m$.
12:      Update the weights: $\boldsymbol{\omega}^{(t+1)} = \boldsymbol{\omega}^{(t)} - \eta_t \left( \mathbf{g} + \lambda \boldsymbol{\omega}^{(t)} \right)$.
13: **end for**
___

# 3 Adaptive Sharpness-Aware Minimization optimizes DNNs under adversarial multiplicative weight perturbations

In this section, we demonstrate that optimizing DNNs with adversarial MWPs follows a similar update rule to Adaptive Sharpness-Aware Minimization (ASAM) (Kwon et al., 2021). We first provide a brief description of ASAM and its predecessor Sharpness-Aware Minimization (SAM) (Foret et al., 2021):

**SAM**    Motivated by previous findings that wide optima tend to generalize better than sharp ones (Keskar et al., 2017; Jiang et al., 2020), SAM regularizes the sharpness of an optimum by solving the following minimax optimization:

$$\min_{\boldsymbol{\omega}} \max_{\|\boldsymbol{\xi}\|_2 \leq \rho} \mathcal{L}(\boldsymbol{\omega} + \boldsymbol{\xi}; \mathcal{S}) + \frac{\lambda}{2} \|\boldsymbol{\omega}\|_F^2 \tag{8}$$

which can be interpreted as optimizing DNNs under adversarial additive weight perturbations. To efficiently solve this problem, Foret et al. (2021) devise a two-step procedure for each iteration $t$:

$$\boldsymbol{\xi}^{(t)} = \rho \frac{\nabla_{\boldsymbol{\omega}} \mathcal{L}(\boldsymbol{\omega}^{(t)}; \mathcal{S})}{\left\|\nabla_{\boldsymbol{\omega}} \mathcal{L}(\boldsymbol{\omega}^{(t)}; \mathcal{S})\right\|_2}, \qquad \boldsymbol{\omega}^{(t+1)} = \boldsymbol{\omega}^{(t)} - \eta_t \left( \nabla_{\boldsymbol{\omega}} \mathcal{L}(\boldsymbol{\omega}^{(t)} + \boldsymbol{\xi}^{(t)}; \mathcal{S}) + \lambda \boldsymbol{\omega}^{(t)} \right) \tag{9}$$

where $\eta_t$ is the learning rate. Each iteration of SAM thus takes twice as long to run than SGD.

**ASAM**    Kwon et al. (2021) note that SAM attempts to minimize the maximum loss over a rigid sphere of radius $\rho$ around an optimum, which is not suitable for ReLU networks since their parameters can be freely re-scaled without affecting the outputs. The authors thus propose ASAM as an alternative optimization problem to SAM which regularizes the *adaptive sharpness* of an optimum:

$$\min_{\boldsymbol{\omega}} \max_{\|T_{\boldsymbol{\omega}}^{-1} \boldsymbol{\xi}\|_2 \leq \rho} \mathcal{L}(\boldsymbol{\omega} + \boldsymbol{\xi}; \mathcal{S}) + \frac{\lambda}{2} \|\boldsymbol{\omega}\|_F^2 \tag{10}$$

where $T_{\boldsymbol{\omega}}$ is an invertible linear operator used to reshape the perturbation region (so that it is not necessarily a sphere as in SAM). Kwon et al. (2021) found that $T_{\boldsymbol{\omega}} = |\boldsymbol{\omega}|$ produced the best results. Solving Eq. (10) in this case leads to the following two-step procedure for each iteration $t$:

$$\widehat{\boldsymbol{\xi}}^{(t)} = \rho \frac{\left(\boldsymbol{\omega}^{(t)}\right)^2 \circ \nabla_{\boldsymbol{\omega}} \mathcal{L}(\boldsymbol{\omega}^{(t)}; \mathcal{S})}{\left\|\boldsymbol{\omega}^{(t)} \circ \nabla_{\boldsymbol{\omega}} \mathcal{L}(\boldsymbol{\omega}^{(t)}; \mathcal{S})\right\|_2}, \quad \boldsymbol{\omega}^{(t+1)} = \boldsymbol{\omega}^{(t)} - \eta_t \left( \nabla_{\boldsymbol{\omega}} \mathcal{L}(\boldsymbol{\omega}^{(t)} + \widehat{\boldsymbol{\xi}}^{(t)}; \mathcal{S}) + \lambda \boldsymbol{\omega}^{(t)} \right) \tag{11}$$

Similar to SAM, each iteration of ASAM also takes twice as long to run than SGD.

**ASAM and adversarial multiplicative perturbations**  Algorithm 1 minimizes the expected loss in Eq. (7). Instead, we could minimize the loss under the adversarial MWP:

$$\mathcal{L}_{\max}(\boldsymbol{\omega}; \mathcal{S}) := \max_{\|\boldsymbol{\xi}\|_2 \leq \rho} \mathcal{L}(\boldsymbol{\omega} + \boldsymbol{\omega} \circ \boldsymbol{\xi}; \mathcal{S}) + \frac{\lambda}{2} \|\boldsymbol{\omega}\|_F^2 \tag{12}$$

Following Foret et al. (2021), we solve this optimization problem by using a first-order Taylor expansion of $\mathcal{L}(\boldsymbol{\omega} + \boldsymbol{\omega} \circ \boldsymbol{\xi}; \mathcal{S})$ to find an approximate solution of the inner maximization:

$$\arg\max_{\|\boldsymbol{\xi}\|_2 \leq \rho} \mathcal{L}(\boldsymbol{\omega} + \boldsymbol{\omega} \circ \boldsymbol{\xi}; \mathcal{S}) \approx \arg\max_{\|\boldsymbol{\xi}\|_2 \leq \rho} \mathcal{L}(\boldsymbol{\omega}; \mathcal{S}) + \langle \boldsymbol{\omega} \circ \boldsymbol{\xi}, \nabla_{\boldsymbol{\omega}} \mathcal{L}(\boldsymbol{\omega}; \mathcal{S}) \rangle \tag{13}$$

The maximizer of the Taylor expansion is:

$$\widehat{\boldsymbol{\xi}}(\boldsymbol{\omega}) = \rho \frac{\boldsymbol{\omega} \circ \nabla_{\boldsymbol{\omega}} \mathcal{L}(\boldsymbol{\omega}; \mathcal{S})}{\|\boldsymbol{\omega} \circ \nabla_{\boldsymbol{\omega}} \mathcal{L}(\boldsymbol{\omega}; \mathcal{S})\|_2} \tag{14}$$

Substituting back into Eq. (12) and differentiating, we get:

$$\nabla_{\boldsymbol{\omega}} \mathcal{L}_{\max}(\boldsymbol{\omega}; \mathcal{S}) \approx \nabla_{\boldsymbol{\omega}} \mathcal{L}(\widehat{\boldsymbol{\omega}}; \mathcal{S}) + \lambda \boldsymbol{\omega} = \nabla_{\boldsymbol{\omega}} \widehat{\boldsymbol{\omega}} \cdot \nabla_{\widehat{\boldsymbol{\omega}}} \mathcal{L}(\widehat{\boldsymbol{\omega}}; \mathcal{S}) + \lambda \boldsymbol{\omega} \tag{15}$$

$$= \nabla_{\widehat{\boldsymbol{\omega}}} \mathcal{L}(\widehat{\boldsymbol{\omega}}; \mathcal{S}) + \nabla_{\boldsymbol{\omega}} \left( \boldsymbol{\omega} \circ \widehat{\boldsymbol{\xi}}(\boldsymbol{\omega}) \right) \cdot \nabla_{\widehat{\boldsymbol{\omega}}} \mathcal{L}(\widehat{\boldsymbol{\omega}}; \mathcal{S}) + \lambda \boldsymbol{\omega} \tag{16}$$

where $\widehat{\boldsymbol{\omega}}$ is the perturbed weight:

$$\widehat{\boldsymbol{\omega}} = \boldsymbol{\omega} + \boldsymbol{\omega} \circ \widehat{\boldsymbol{\xi}}(\boldsymbol{\omega}) = \boldsymbol{\omega} + \rho \frac{\boldsymbol{\omega}^2 \circ \nabla_{\boldsymbol{\omega}} \mathcal{L}(\boldsymbol{\omega}; \mathcal{S})}{\|\boldsymbol{\omega} \circ \nabla_{\boldsymbol{\omega}} \mathcal{L}(\boldsymbol{\omega}; \mathcal{S})\|_2} \tag{17}$$

Similar to Foret et al. (2021), we omit the second summand in Eq. (16) for efficiency, as it requires calculating the Hessian of the loss. We then arrive at the gradient formula in the update rule of ASAM in Eq. (11). We have thus established a connection between ASAM and adversarial MWPs.

# 4  Empirical evaluation

In this section, we assess the corruption robustness of DAMP and ASAM in image classification tasks. We conduct experiments using the CIFAR-10/100 (Krizhevsky, 2009), TinyImageNet (Le and Yang, 2015), and ImageNet (Deng et al., 2009) datasets. For evaluation on corrupted images, we utilize the CIFAR-10/100-C, TinyImageNet-C, and ImageNet-C datasets provided by Hendrycks and Dietterich (2019), as well as ImageNet-$\overline{\text{C}}$ (Mintun et al., 2021), ImageNet-D (Zhang et al., 2024), ImageNet-A (Hendrycks et al., 2021), ImageNet-Sketch (Wang et al., 2019), ImageNet-{Drawing, Cartoon} (Salvador and Oberman, 2022), and ImageNet-Hard (Taesiri et al., 2023) datasets, which encapsulate a wide range of corruptions. Detail descriptions of these datasets are provided in Appendix E. We further evaluate the models on adversarial examples generated by the Fast Gradient Sign Method (FGSM) (Goodfellow et al., 2014). In terms of architectures, we use ResNet18 (He et al., 2016a) for CIFAR-10/100, PreActResNet18 (He et al., 2016b) for TinyImageNet, ResNet50 (He et al., 2016a), ViT-S/16, and ViT-B/16 (Dosovitskiy et al., 2021) for ImageNet. We ran all experiments on a single machine with 8 Nvidia V100 GPUs. Appendix F includes detailed information for each experiment.

## 4.1  Comparing DAMP to directly using corruptions as augmentations

In this section, we compare the corruption robustness of DNNs trained using DAMP with those trained directly on corrupted images. To train models on corrupted images, we utilize Algorithm 2 described in the Appendix. For a given target corruption **g**, Algorithm 2 randomly selects half the images in each training batch and applies **g** to them. This random selection process enhances the final model's robustness to the target corruption while maintaining its accuracy on clean images. We use the `imagecorruptions` library (Michaelis et al., 2019) to apply the corruptions during training.

**Evaluation metric**  We use the corruption error $\text{CE}_c^f$ (Hendrycks and Dietterich, 2019) which measures the predictive error of classifier $f$ in the presence of corruption $c$. Denote $E_{s,c}^f$ as the error of classifier $f$ under corruption $c$ with corruption severity $s$, the corruption error $\text{CE}_c^f$ is defined as:

$$\text{CE}_c^f = \left( \sum_{s=1}^5 E_{s,c}^f \right) \Big/ \left( \sum_{s=1}^5 E_{s,c}^{f_{\text{baseline}}} \right) \tag{18}$$

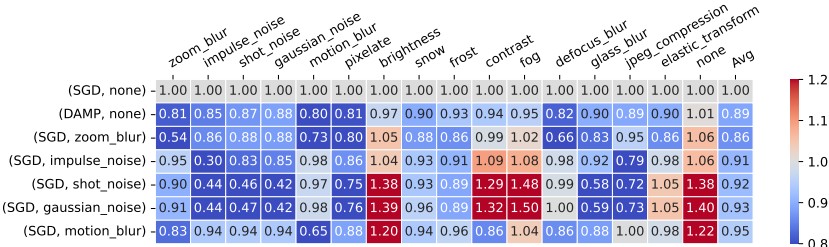

Figure 3: **DAMP improves robustness to all corruptions while preserving accuracy on clean images.** Results of ResNet18/CIFAR-100 experiments averaged over 5 seeds. The heatmap shows $\text{CE}_c^f$ described in Eq. (18) (lower is better), where each row corresponds to a tuple of training (`method`, `corruption`), while each column corresponds to the test corruption. The `Avg` column shows the average of the results of the previous columns. `none` indicates no corruption. We use the models trained under the `SGD/none` setting (first row) as baselines to calculate the $\text{CE}_c^f$. The last five rows are the 5 best training corruptions ranked by the results in the `Avg` column.

For this metric, lower is better. Here $f_{\text{baseline}}$ is a baseline classifier whose usage is to make the error more comparable between corruptions as some corruptions can be more challenging than others (Hendrycks and Dietterich, 2019). For each experiment setting, we use the model trained by SGD without corruptions as $f_{\text{baseline}}$.

**Results**    We visualize the results for the ResNet18/CIFAR-100 setting in Fig. 3. The results for the ResNet18/CIFAR-10 and PreActResNet18/TinyImageNet settings are presented in Figs. 5 and 6 in the Appendix. Figs. 3, 5 and 6 demonstrate that DAMP improves predictive accuracy over plain SGD across all corruptions without compromising accuracy on clean images. Although Fig. 3 indicates that including `zoom_blur` as an augmentation when training ResNet18 on CIFAR-100 yields better results than DAMP on average, it also reduces accuracy on clean images and the `brightness` corruption. Overall, these figures show that incorporating a specific corruption as data augmentation during training enhances robustness to that particular corruption but may reduce performance on clean images and other corruptions. In contrast, DAMP consistently improves robustness across all corruptions. Notably, DAMP even enhances accuracy on clean images in the PreActResNet18/TinyImageNet setting, as shown in Fig. 6.

### 4.2    Comparing DAMP to random additive perturbations

In this section, we investigate whether additive weight perturbations can also enhance corruption robustness. To this end, we compare DAMP with its variant, Data Augmentation via Additive Perturbations (DAAP). Unlike DAMP, DAAP perturbs weights during training with random additive Gaussian noises centered at 0, as detailed in Algorithm 3 in the Appendix. Fig. 7 in the Appendix presents the results of DAMP and DAAP under different noise standard deviations, alongside standard SGD. Overall, Fig. 7 shows that across different experimental settings, the corruption robustness of DAAP is only slightly better than SGD and is worse than DAMP. Therefore, we conclude that MWPs are better than their additive counterparts at improving robustness to corruptions.

### 4.3    Benchmark results

In this section, we compare DAMP with Dropout (Srivastava et al., 2014), SAM (Foret et al., 2021), and ASAM (Kwon et al., 2021). For SAM and ASAM, we optimize the neighborhood size $\rho$ by using 10% of the training set as a validation set. Similarly, we adjust the noise standard deviation $\sigma$ for DAMP and the drop rate $p$ for Dropout following the same procedure. For hyperparameters and additional training details, please refer to Appendix F.

**CIFAR-10/100 and TinyImageNet.**    Fig. 4 visualizes the predictive errors of DAMP and the baseline methods on CIFAR-10/100 and TinyImageNet, with all methods trained for the same number of epochs. It demonstrates that DAMP consistently outperforms Dropout across various datasets and corruption severities, despite having the same training cost. Notably, DAMP outperforms SAM under

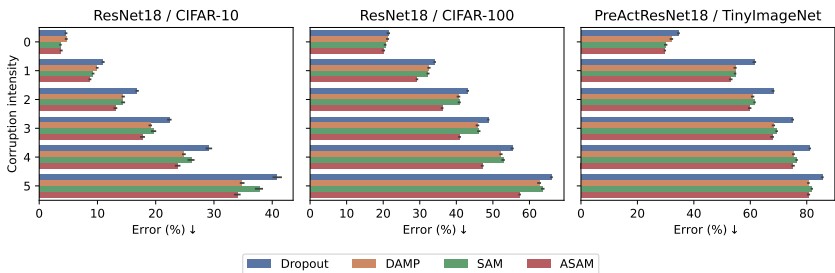

Figure 4: **DAMP surpasses SAM on corrupted images in most cases, despite requiring only half the training cost.** We report the predictive errors (lower is better) averaged over 5 seeds. A severity level of 0 indicates no corruption. We use the same number of epochs for all methods.

Table 1: **DAMP surpasses the baselines on corrupted images in most cases and on average.** We report the predictive errors (lower is better) averaged over 3 seeds for the ResNet50 / ImageNet experiments. Subscript numbers represent standard deviations. We evaluate the models on IN-{C, $\overline{\text{C}}$, A, D, Sketch, Drawing, Cartoon, Hard}, and adversarial examples generated by FGSM. For FGSM, we use $\epsilon = 2/224$. For IN-{C, $\overline{\text{C}}$}, we report the results averaged over all corruption types and severity levels. We use 90 epochs and the basic Inception-style preprocessing for all experiments.

| Method | Clean Error (%) ↓ | Corrupted Error (%) ↓ | | | | | | | | | |
| --- | --- | --- | --- | --- | --- | --- | --- | --- | --- | --- | --- |
| | | FGSM | A | C | $\overline{\text{C}}$ | Cartoon | D | Drawing | Sketch | Hard | Avg |
| Dropout | $23.6_{0.2}$ | $90.7_{0.2}$ | $\mathbf{95.7}_{<0.1}$ | $61.7_{0.2}$ | $61.6_{<0.1}$ | $49.6_{0.2}$ | $88.9_{<0.1}$ | $77.4_{1.3}$ | $78.3_{0.3}$ | $85.8_{0.1}$ | $76.6$ |
| DAMP | $23.8_{<0.1}$ | $\mathbf{88.3}_{0.1}$ | $96.2_{<0.1}$ | $\mathbf{58.6}_{0.1}$ | $\mathbf{58.7}_{<0.1}$ | $\mathbf{44.4}_{<0.1}$ | $88.7_{<0.1}$ | $\mathbf{71.1}_{0.5}$ | $\mathbf{76.3}_{0.2}$ | $85.3_{0.2}$ | $\mathbf{74.2}$ |
| SAM | $23.2_{<0.1}$ | $90.4_{0.2}$ | $96.6_{0.1}$ | $60.2_{0.1}$ | $60.7_{0.1}$ | $47.6_{0.1}$ | $\mathbf{88.3}_{0.1}$ | $74.8_{<0.1}$ | $77.5_{0.1}$ | $85.8_{0.3}$ | $75.8$ |
| ASAM | $\mathbf{22.8}_{0.1}$ | $89.7_{0.2}$ | $96.8_{0.1}$ | $58.9_{0.1}$ | $59.2_{0.1}$ | $45.5_{<0.1}$ | $88.7_{0.1}$ | $72.3_{0.1}$ | $76.4_{0.2}$ | $\mathbf{85.2}_{0.1}$ | $74.7$ |

most corruption scenarios, even though SAM takes twice as long to train and has higher accuracy on clean images. Additionally, DAMP improves accuracy on clean images over Dropout on CIFAR-100 and TinyImageNet. Finally, ASAM consistently surpasses other methods on both clean and corrupted images, as it employs adversarial MWPs (Section 3). However, like SAM, each ASAM experiment takes twice as long as DAMP given the same epoch counts.

**ResNet50 / ImageNet**    Table 1 presents the predictive errors for the ResNet50 / ImageNet setting on a variety of corruption test sets. It shows that DAMP consistently outperforms the baselines in most corruption scenarios and on average, despite having half the training cost of SAM and ASAM.

**ViT-S16 / ImageNet / Basic augmentations**    Table 2 presents the predictive errors for the ViT-S16 / ImageNet setting, using the training setup from Beyer et al. (2022) but with only basic Inception-style preprocessing (Szegedy et al., 2016). Remarkably, DAMP can train ViT-S16 from scratch in 200 epochs to match ResNet50's accuracy without advanced data augmentation. This is significant as ViT typically requires either extensive pretraining (Dosovitskiy et al., 2021), comprehensive data augmentation (Beyer et al., 2022), sophisticated training techniques (Chen et al., 2022), or modifications to the original architecture (Yuan et al., 2021) to perform well on ImageNet. Additionally, DAMP consistently ranks in the top 2 for corruption robustness across various test settings and has the best corruption robustness on average (last column). Comparing Tables 1 and 2 reveals that ViT-S16 is more robust to corruptions than ResNet50 when both have similar performance on clean images.

**ViT / ImageNet / Advanced augmentations**    Table 3 presents the predictive errors of ViT-S16 and ViT-B16 on ImageNet with MixUp (Zhang et al., 2018) and RandAugment (Cubuk et al., 2020). These results indicate that DAMP can be combined with modern augmentation techniques to further improve robustness. Furthermore, using DAMP to train a larger model (ViT-B16) yields better results than using SAM/ASAM to train a smaller model (ViT-S16), given the same amount of training time.

Table 2: **ViT-S16 / ImageNet (IN) with basic Inception-style data augmentations**. Due to the high training cost, we report the predictive error (lower is better) of a single run. We evaluate corruption robustness of the models using IN-{C, $\overline{C}$, A, D, Sketch, Drawing, Cartoon, Hard}, and adversarial examples generated by FGSM. For IN-{C, $\overline{C}$}, we report the results averaged over all corruption types and severity levels. For FGSM, we use $\epsilon = 2/224$. We also report the runtime of each experiment, showing that SAM and ASAM take twice as long to run than DAMP and AdamW given the same number of epochs. DAMP produces the most robust model on average.

| Method | #Epochs | Runtime | Clean Error (%)↓ | Corrupted Error (%)↓ | | | | | | | | | |
| --- | --- | --- | --- | --- | --- | --- | --- | --- | --- | --- | --- | --- | --- |
| | | | | FGSM | A | C | $\overline{C}$ | Cartoon | D | Drawing | Sketch | Hard | Avg |
| Dropout | 100 | 20.6h | 28.55 | 93.47 | 93.44 | 65.87 | 64.52 | 50.37 | 91.15 | 79.62 | 88.06 | 87.19 | 79.30 |
| | 200 | 41.1h | 28.74 | 90.95 | 93.33 | 66.90 | 64.83 | 51.23 | 92.56 | 81.24 | 87.99 | 87.60 | 79.63 |
| DAMP | 100 | 20.7h | 25.50 | 92.76 | 92.92 | 57.85 | 57.02 | 44.78 | 88.79 | 69.92 | 83.16 | 85.65 | 74.76 |
| | 200 | 41.1h | **23.75** | **84.33** | **90.56** | 55.58 | **55.58** | 41.06 | **87.87** | 68.36 | 81.82 | **84.18** | **72.15** |
| SAM | 100 | 41h | 23.91 | 87.61 | 93.96 | 55.56 | 55.93 | 42.53 | 88.23 | 69.53 | 81.86 | 85.54 | 73.42 |
| ASAM | 100 | 41.1h | 24.01 | 85.85 | 92.99 | **55.13** | 55.64 | **40.74** | 89.03 | **67.80** | **81.47** | 84.31 | 72.55 |

Table 3: **ViT / ImageNet (IN) with MixUp and RandAugment**. We train ViT-S16 and ViT-B16 on ImageNet from scratch with advanced data augmentations (DAs). We evaluate the models on IN-{C, $\overline{C}$, A, D, Sketch, Drawing, Cartoon, Hard}, and adversarial examples generated by FGSM. For FGSM, we use $\epsilon = 2/224$. For IN-{C, $\overline{C}$}, we report the results averaged over all corruption types and severity levels. These results indicate that: (i) DAMP can be combined with modern DA techniques to further enhance robustness; (ii) DAMP is capable of training large models like ViT-B16; (iii) given the same amount of training time, it is better to train a large model (ViT-B16) using DAMP than to train a smaller model (ViT-S16) using SAM/ASAM.

| Model | Method | #Epochs | Runtime | Clean Error (%)↓ | Corrupted Error (%)↓ | | | | | | | | | |
| --- | --- | --- | --- | --- | --- | --- | --- | --- | --- | --- | --- | --- | --- | --- |
| | | | | | FGSM | $\overline{C}$ | A | C | Cartoon | D | Drawing | Sketch | Hard | Avg |
| ViT S16 | Dropout | 500 | 111h | 20.25 | 62.45 | 40.85 | 84.29 | 44.72 | 34.35 | 86.59 | 56.31 | 71.03 | 80.87 | 62.38 |
| | DAMP | 500 | 111h | **20.09** | 59.87 | **39.30** | **83.12** | **43.18** | 34.01 | **84.74** | 54.16 | **68.03** | **80.05** | **60.72** |
| | SAM | 300 | 123h | 20.17 | 59.92 | 40.05 | 83.91 | 44.04 | 34.34 | 85.99 | 55.63 | 70.85 | 80.18 | 61.66 |
| | ASAM | 300 | 123h | 20.38 | **59.38** | 39.44 | 83.64 | 43.41 | **33.82** | 85.41 | 54.43 | 69.13 | 80.50 | 61.02 |
| ViT B16 | Dropout | 275 | 123h | 20.41 | 56.43 | 39.14 | 82.85 | 43.82 | 33.13 | 87.72 | 56.15 | 71.36 | 79.13 | 61.08 |
| | DAMP | 275 | 124h | **19.36** | 55.20 | 37.77 | 80.49 | 41.67 | 31.63 | 87.06 | 52.32 | **67.91** | **78.69** | **59.19** |
| | SAM | 150 | 135h | 19.84 | 61.85 | 39.09 | 82.69 | 43.53 | 32.95 | 88.38 | 55.33 | 71.22 | 79.48 | 61.61 |
| | ASAM | 150 | 136h | 19.40 | 58.87 | **37.41** | 82.21 | **41.18** | **30.76** | 88.03 | **51.84** | 69.54 | 78.83 | 59.85 |

## 5   Related works

**Dropout**   Perhaps most relevant to our method is Dropout (Srivastava et al., 2014) and its many variants, such as DropConnect (Wan et al., 2013) and Variational Dropout (Kingma et al., 2015). These methods can be viewed as DAMP where the noise distribution $\Xi$ is a structured multivariate Bernoulli distribution. For instance, Dropout multiplies all the weights connecting to a node with a binary random variable $p \sim \text{Bernoulli}(\rho)$. While the main motivation of these Dropout methods is to prevent co-adaptations of neurons to improve generalization on clean data, the motivation of DAMP is to improve robustness to input corruptions without harming accuracy on clean data. Nonetheless, our experiments show that DAMP can improve generalization on clean data in certain scenarios, such as PreActResNet18/TinyImageNet and ViT-S16/ImageNet.

**Ensemble methods**   Ensemble methods, such as Deep ensembles (Lakshminarayanan et al., 2017) and Bayesian neural networks (BNNs) (Graves, 2011; Blundell et al., 2015; Gal and Ghahramani, 2016; Louizos and Welling, 2017; Izmailov et al., 2021; Trinh et al., 2022), have been explored as effective defenses against corruptions. Ovadia et al. (2019) benchmarked some of these methods, demonstrating that they are more robust to corruptions compared to a single model. However, the training and inference costs of these methods increase linearly with the number of ensemble members, making them inefficient for use with very large DNNs.

**Data augmentation**   Data augmentations aim at enhancing robustness include AugMix (Hendrycks et al., 2019), which combines common image transformations; Patch Gaussian (Lopes et al., 2019), which applies Gaussian noise to square patches; ANT (Rusak et al., 2020), which uses adversarially learned noise distributions for augmentation; and AutoAugment (Cubuk et al., 2018), which learns

augmentation policies directly from the training data. These methods have been demonstrated to improve robustness to the corruptions in ImageNet-C (Hendrycks and Dietterich, 2019). Mintun et al. (2021) attribute the success of these methods to the fact that they generate augmented images perceptually similar to the corruptions in ImageNet-C and propose ImageNet-C̄, a test set of 10 new corruptions that are challenging to models trained by these augmentation methods.

**Test-time adaptations via BatchNorm**  One effective approach to using unlabelled data to improve corruption robustness is to keep BatchNorm (Ioffe and Szegedy, 2015) on at test-time to adapt the batch statistics to the corrupted test data (Li et al., 2016; Nado et al., 2020; Schneider et al., 2020; Benz et al., 2021). A major drawback is that this approach cannot be used with BatchNorm-free architectures, such as Vision Transformer (Dosovitskiy et al., 2021).

## 6   Conclusion

In this work, we demonstrate that MWPs improve robustness of DNNs to a wide range of input corruptions. We introduce DAMP, a simple training algorithm that perturbs weights during training with random multiplicative noise while maintaining the same training cost as standard SGD. We further show that ASAM (Kwon et al., 2021) can be viewed as optimizing DNNs under adversarial MWPs. Our experiments show that both DAMP and ASAM indeed produce models that are robust to corruptions. DAMP is also shown to improve sample efficiency of Vision Transformer, allowing it to achieve comparable performance to ResNet50 on medium size datasets such as ImageNet without extensive data augmentations. Additionally, DAMP can be used in conjunction with modern augmentation techniques such as MixUp and RandAugment to further boost robustness. As DAMP is domain-agnostic, one future direction is to explore its effectiveness in domains other than computer vision, such as natural language processing and reinforcement learning. Another direction is to explore alternative noise distributions to the Gaussian distribution used in our work.

**Limitations**  Here we outline some limitations of this work. First, the proof of Theorem 1 assumes a simple feedforward neural network, thus it does not take into accounts modern DNN's components such as normalization layers and attentions. Second, we only explored random Gaussian multiplicative perturbations, and there are likely more sophisticated noise distributions that could further boost corruption robustness.

## Broader Impacts

Our paper introduces a new training method for neural networks that improves their robustness to input corruptions. Therefore, we believe that our work contributes towards making deep leading models safer and more reliable to use in real-world applications, especially those that are safety-critical. However, as with other methods that improve robustness, our method could also be improperly used in applications that negatively impact society, such as making mass surveillance systems more accurate and harder to fool. To this end, we hope that practitioners carefully consider issues regarding fairness, bias and other potentially harmful societal impacts when designing deep learning applications.

## Acknowledgments

This work was supported by the Research Council of Finland (Flagship programme: Finnish Center for Artificial Intelligence FCAI and decision no. 359567, 345604 and 341763), ELISE Networks of Excellence Centres (EU Horizon: 2020 grant agreement 951847) and UKRI Turing AI World-Leading Researcher Fellowship (EP/W002973/1). We acknowledge the computational resources provided by Aalto Science-IT project and CSC-IT Center for Science, Finland.

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

# A Proof of Lemma 1

*Proof.* Here we note that:

$$\mathbf{x}^{(h)} := \mathbf{f}^{(h)}(\mathbf{x}) \tag{19}$$

$$\mathbf{x}_{\mathbf{g}}^{(h)} := \mathbf{f}^{(h)}(\mathbf{g}(\mathbf{x})) \tag{20}$$

$$\boldsymbol{\delta}_{\mathbf{g}}\ell(\boldsymbol{\omega}, \mathbf{x}, y) := \ell(\boldsymbol{\omega}, \mathbf{x}_{\mathbf{g}}, y) - \ell(\boldsymbol{\omega}, \mathbf{x}, y) \tag{21}$$

$$\boldsymbol{\delta}_{\mathbf{g}}\mathbf{x}^{(h)} := \mathbf{x}_{\mathbf{g}}^{(h)} - \mathbf{x}^{(h)} \tag{22}$$

We first notice that the per-sample loss $\ell(\boldsymbol{\omega}, \mathbf{x}, y)$ can be viewed as a function of the intermediate activation $\mathbf{x}^{(h)}$ of layer $h$ (see Fig. 2). From Assumption 3, there exists a constant $L_h > 0$ such that:

$$\|\nabla_{\mathbf{x}_{\mathbf{g}}^{(h)}}\ell(\boldsymbol{\omega}, \mathbf{x_g}, y) - \nabla_{\mathbf{x}^{(h)}}\ell(\boldsymbol{\omega}, \mathbf{x}, y)\|_2 \leq L_h \|\boldsymbol{\delta}_{\mathbf{g}}\mathbf{x}^{(h)}\|_2 \tag{23}$$

which gives us the following quadratic bound:

$$\ell(\boldsymbol{\omega}, \mathbf{x_g}, y) \leq \ell(\boldsymbol{\omega}, \mathbf{x}, y) + \left\langle \nabla_{\mathbf{x}^{(h)}}\ell(\boldsymbol{\omega}, \mathbf{x}, y), \boldsymbol{\delta}_{\mathbf{g}}\mathbf{x}^{(h)} \right\rangle + \frac{L_h}{2}\|\boldsymbol{\delta}_{\mathbf{g}}\mathbf{x}^{(h)}\|_2^2 \tag{24}$$

where $\langle \cdot, \cdot \rangle$ denotes the dot product between two vectors. The results in the equation above have been proven in Böhning and Lindsay (1988). Subtracting $\ell(\boldsymbol{\omega}, \mathbf{x}, y)$ from both side of Eq. (24) gives us:

$$\boldsymbol{\delta}_{\mathbf{g}}\ell(\boldsymbol{\omega}, \mathbf{x}, y) \leq \left\langle \nabla_{\mathbf{x}^{(h)}}\ell(\boldsymbol{\omega}, \mathbf{x}, y), \boldsymbol{\delta}_{\mathbf{g}}\mathbf{x}^{(h)} \right\rangle + \frac{L_h}{2}\|\boldsymbol{\delta}_{\mathbf{g}}\mathbf{x}^{(h)}\|_2^2 \tag{25}$$

Since the pre-activation output of layer $h + 1$ is $\mathbf{z}^{(h+1)}(\mathbf{x}) = \mathbf{W}^{(h+1)}\mathbf{f}^{(h)}(\mathbf{x}) = \mathbf{W}^{(h+1)}\mathbf{x}^{(h)}$, we can rewrite the inequality above as:

$$\boldsymbol{\delta}_{\mathbf{g}}\ell(\boldsymbol{\omega}, \mathbf{x}, y) \leq \left\langle \nabla_{\mathbf{z}^{(h+1)}}\ell(\boldsymbol{\omega}, \mathbf{x}, y) \otimes \boldsymbol{\delta}_{\mathbf{g}}\mathbf{x}^{(h)}, \mathbf{W}^{(h+1)} \right\rangle_F + \frac{L_h}{2}\|\boldsymbol{\delta}_{\mathbf{g}}\mathbf{x}^{(h)}\|_2^2 \tag{26}$$

where $\otimes$ denotes the outer product of two vectors and $\langle \cdot, \cdot \rangle_F$ denotes the Frobenius inner product of two matrices of similar dimension.

From Assumption 1, we have that there exists a constant $M > 0$ such that:

$$\|\boldsymbol{\delta}_{\mathbf{g}}\mathbf{x}^{(0)}\|_2^2 = \|\mathbf{x}_{\mathbf{g}}^{(0)} - \mathbf{x}^{(0)}\|_2^2 = \|\mathbf{g}(\mathbf{x}) - \mathbf{x}\|_2^2 \leq M \tag{27}$$

Given that $\mathbf{x}^{(1)} = \boldsymbol{\sigma}^{(1)}\left(\mathbf{W}^{(1)}\mathbf{x}^{(0)}\right)$, we have:

$$\|\boldsymbol{\delta}_{\mathbf{g}}\mathbf{x}^{(1)}\|_2^2 = \|\mathbf{x}_{\mathbf{g}}^{(1)} - \mathbf{x}^{(1)}\|_2^2 \leq \|\mathbf{W}^{(1)}\boldsymbol{\delta}_{\mathbf{g}}\mathbf{x}^{(0)}\|_2^2 \tag{28}$$

Here we assume that the activate $\boldsymbol{\sigma}$ satisfies $\|\boldsymbol{\sigma}(\mathbf{x}) - \boldsymbol{\sigma}(\mathbf{y})\|_2 \leq \|\mathbf{x} - \mathbf{y}\|_2$, which is true for modern activation functions such as ReLU. Since $\|\boldsymbol{\delta}_{\mathbf{g}}\mathbf{x}^{(0)}\|_2^2$ is bounded, there exists a constant $\hat{C}_{\mathbf{g}}^{(1)}(\mathbf{x})$ such that:

$$\|\boldsymbol{\delta}_{\mathbf{g}}\mathbf{x}^{(1)}\|_2^2 = \|\mathbf{x}_{\mathbf{g}}^{(1)} - \mathbf{x}^{(1)}\|_2^2 \leq \|\mathbf{W}^{(1)}\boldsymbol{\delta}_{\mathbf{g}}\mathbf{x}^{(0)}\|_2^2 \leq \frac{\hat{C}_{\mathbf{g}}^{(1)}(\mathbf{x})}{2}\|\mathbf{W}^{(1)}\|_F^2 \tag{29}$$

where $\|\cdot\|_F$ denotes the Frobenius norm. Similarly, as we have proven that $\|\boldsymbol{\delta}_{\mathbf{g}}\mathbf{x}^{(1)}\|_2^2$ is bounded, there exists a constant $\hat{C}_{\mathbf{g}}^{(2)}(\mathbf{x})$ such that:

$$\|\boldsymbol{\delta}_{\mathbf{g}}\mathbf{x}^{(2)}\|_2^2 = \|\mathbf{x}_{\mathbf{g}}^{(2)} - \mathbf{x}^{(2)}\|_2^2 \leq \|\mathbf{W}^{(2)}\boldsymbol{\delta}_{\mathbf{g}}\mathbf{x}^{(1)}\|_2^2 \leq \frac{\hat{C}_{\mathbf{g}}^{(2)}(\mathbf{x})}{2}\|\mathbf{W}^{(2)}\|_F^2 \tag{30}$$

Thus we have proven that for all $h = 1, \ldots, H$, there exists a constant $\hat{C}_{\mathbf{g}}^{(h)}(\mathbf{x})$ such that:

$$\|\boldsymbol{\delta}_{\mathbf{g}}\mathbf{x}^{(h)}\|_2^2 \leq \frac{\hat{C}_{\mathbf{g}}^{(h)}(\mathbf{x})}{2}\|\mathbf{W}^{(h)}\|_F^2 \tag{31}$$

By combining Eqs. (26) and (31) and setting $C_{\mathbf{g}}^{(h)}(\mathbf{x}) = L_h \hat{C}_{\mathbf{g}}^{(h)}(\mathbf{x})$, we arrive at Eq. (4). □

# B   Proof of Theorem 1

*Proof.* From Lemma 1, we have for all $h = 0, \ldots, H - 1$:

$$\mathcal{L}(\boldsymbol{\omega}; \mathbf{g}(\mathcal{S})) = \frac{1}{N} \sum_{k=1}^{N} \ell(\boldsymbol{\omega}, \mathbf{g}(\mathbf{x}_k), y_k) = \frac{1}{N} \sum_{k=1}^{N} \left( \ell(\boldsymbol{\omega}, \mathbf{x}_k, y_k) + \boldsymbol{\delta}_{\mathbf{g}} \ell(\boldsymbol{\omega}, \mathbf{x}_k, y_k) \right) \tag{32}$$

$$\leq \mathcal{L}(\boldsymbol{\omega}; \mathcal{S}) + \frac{1}{N} \sum_{k=1}^{N} \left\langle \nabla_{\mathbf{z}^{(h+1)}} \ell(\boldsymbol{\omega}, \mathbf{x}_k, y_k) \otimes \boldsymbol{\delta}_{\mathbf{g}} \mathbf{x}_k^{(h)}, \mathbf{W}^{(h+1)} \right\rangle_F + \frac{\hat{C}_{\mathbf{g}}^{(h)}}{2} \|\mathbf{W}^{(h)}\|_F^2 \tag{33}$$

where $\hat{C}_{\mathbf{g}}^{(h)} = \max_{\mathbf{x} \in \mathcal{S}} C_{\mathbf{g}}^{(h)}(\mathbf{x})$. Since this bound is true for all $h$, we can take the average:

$$\mathcal{L}(\boldsymbol{\omega}; \mathbf{g}(\mathcal{S})) \leq \mathcal{L}(\boldsymbol{\omega}; \mathcal{S}) + \frac{1}{H} \sum_{h=1}^{H} \frac{1}{N} \sum_{k=1}^{N} \left\langle \nabla_{\mathbf{z}^{(h)}} \ell(\boldsymbol{\omega}, \mathbf{x}_k, y_k) \otimes \boldsymbol{\delta}_{\mathbf{g}} \mathbf{x}_k^{(h-1)}, \mathbf{W}^{(h)} \right\rangle_F$$
$$+ \frac{C_{\mathbf{g}}}{2} \|\boldsymbol{\omega}\|_F^2 \tag{34}$$

where $C_{\mathbf{g}} = \frac{1}{H} \sum_{h=1}^{H} \hat{C}_{\mathbf{g}}^{(h)}$. The right-hand side of Eq. (34) can be written as:

$$\mathcal{L}(\boldsymbol{\omega}; \mathcal{S}) + \frac{1}{H} \sum_{h=1}^{H} \left\langle \frac{1}{N} \sum_{k=1}^{N} \nabla_{\mathbf{z}^{(h)}} \ell(\boldsymbol{\omega}, \mathbf{x}_k, y_k) \otimes \boldsymbol{\delta}_{\mathbf{g}} \mathbf{x}_k^{(h-1)}, \mathbf{W}^{(h)} \right\rangle_F + \frac{C_{\mathbf{g}}}{2} \|\boldsymbol{\omega}\|_F^2 \tag{35}$$

$$= \mathcal{L}(\boldsymbol{\omega}; \mathcal{S}) + \sum_{h=1}^{H} \left\langle \nabla_{\mathbf{W}^{(h)}} \mathcal{L}(\boldsymbol{\omega}; \mathcal{S}), \mathbf{W}^{(h)} \circ \boldsymbol{\xi}^{(h)}(\mathbf{g}) \right\rangle_F + \frac{C_{\mathbf{g}}}{2} \|\boldsymbol{\omega}\|_F^2 \tag{36}$$

$$\leq \mathcal{L}(\boldsymbol{\omega} + \boldsymbol{\omega} \circ \boldsymbol{\xi}(\mathbf{g}); \mathcal{S}) + \frac{C_{\mathbf{g}}}{2} \|\boldsymbol{\omega}\|_F^2 = \mathcal{L}(\boldsymbol{\omega} \circ (1 + \boldsymbol{\xi}(\mathbf{g})); \mathcal{S}) + \frac{C_{\mathbf{g}}}{2} \|\boldsymbol{\omega}\|_F^2 \tag{37}$$

where $\boldsymbol{\xi}^{(h)}(\mathbf{g})$ is a matrix of the same dimension as $\mathbf{W}^{(h)}$ whose each entry is defined as:

$$\left[ \boldsymbol{\xi}^{(h)}(\mathbf{g}) \right]_{i,j} = \frac{1}{H} \frac{\left[ \sum_{k=1}^{N} \nabla_{\mathbf{z}^{(h)}} \ell(\boldsymbol{\omega}, \mathbf{x}_k, y_k) \otimes \boldsymbol{\delta}_{\mathbf{g}} \mathbf{x}_k^{(h-1)} \right]_{i,j}}{\left[ \sum_{k=1}^{N} \nabla_{\mathbf{z}^{(h)}} \ell(\boldsymbol{\omega}, \mathbf{x}_k, y_k) \otimes \mathbf{x}_k^{(h-1)} \right]_{i,j}} \tag{38}$$

The inequality in Eq. (37) is due to the first-order Taylor expansion and the assumption that the training loss is locally convex at $\boldsymbol{\omega}$. This assumption is expected to hold for the final solution but does not necessarily hold for any $\boldsymbol{\omega}$. Eq. (5) is obtained by combining Eq. (34) and Eq. (37). $\qquad\square$

# C   Training with corruption

Here we present Algorithm 2 which uses corruptions as data augmentation during training, as well as the experiment results of Section 4.1 for ResNet18/CIFAR-10 and PreActResNet18/TinyImageNet settings in Figs. 5 and 6.

# D   Training with random additive weight perturbations

Here, we present Algorithm 3 used in Section 4.2 which trains DNNs under random additive weight perturbations and Fig. 7 comparing performance between DAMP and DAAP.

# E   Corruption datasets

**CIFAR-10/100-C (Hendrycks and Dietterich, 2019)**   These datasets contain the corrupted versions of the CIFAR-10/100 test sets. They contain 19 types of corruption, each divided into 5 levels of severity.

**TinyImageNet-C (Hendrycks and Dietterich, 2019)**   This dataset contains the corrupted versions of the TinyImageNet test set. It contains 19 types of corruption, each divided into 5 levels of severity.

---

**Algorithm 2** Training with corruption

---

1: **Input:** training data $\mathcal{S} = \{(\mathbf{x}_k, y_k)\}_{k=1}^N$, a neural network $\mathbf{f}(\cdot; \boldsymbol{\omega})$ parameterized by $\boldsymbol{\omega} \in \mathbb{R}^P$, number of iterations $T$, step sizes $\{\eta_t\}_{t=1}^T$, batch size $B$, a corruption $\mathbf{g}$ such as Gaussian noise, weight decay coefficient $\lambda$, a loss function $\mathcal{L} : \mathbb{R}^P \to \mathbb{R}_+$.
2: **Output:** Optimized parameter $\boldsymbol{\omega}^{(T)}$.t
3: Initialize parameter $\boldsymbol{\omega}^{(0)}$.
4: **for** $t = 1$ **to** $T$ **do**
5:     Draw a mini-batch $\mathcal{B} = \{(\mathbf{x}_b, y_b)\}_{b=1}^B \sim \mathcal{S}$.
6:     Divide the mini-batch into two disjoint sub-batches of equal size $\mathcal{B}_1$ and $\mathcal{B}_2$.
7:     Apply the corruption $\mathbf{g}$ to all samples in $\mathcal{B}_1$: $\mathbf{g}(\mathcal{B}_1) = \{(\mathbf{g}(\mathbf{x}), y)\}_{(\mathbf{x},y) \in \mathcal{B}_1}$.
8:     Compute the gradient $\mathbf{g} = \nabla_{\boldsymbol{\omega}} \mathcal{L}(\boldsymbol{\omega}; \mathbf{g}(\mathcal{B}_1) \cup \mathcal{B}_2)$.
9:     Update the weights: $\boldsymbol{\omega}^{(t+1)} = \boldsymbol{\omega}^{(t)} - \eta_t \left( \mathbf{g} + \lambda \boldsymbol{\omega}^{(t)} \right)$.
10: **end for**

---

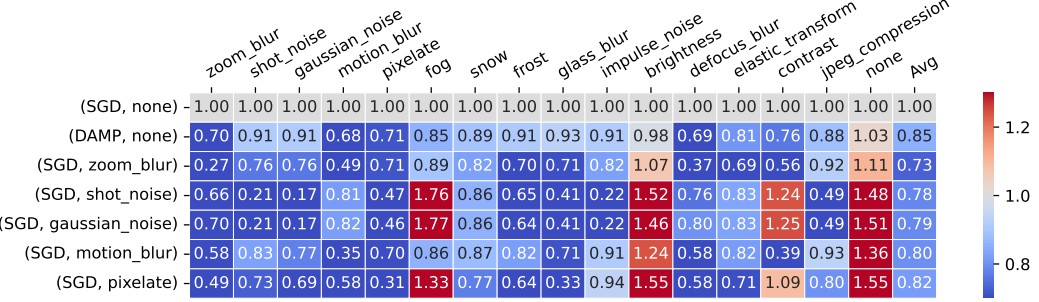

Figure 5: **DAMP improves robustness to all corruptions while preserving accuracy on clean images.** Results of ResNet18/CIFAR-10 experiments averaged over 3 seeds. The heatmap shows $\mathrm{CE}_c^f$ described in Eq. (18), where each row corresponds to a tuple of of training (`method`, `corruption`), while each column corresponds to the test corruption. The `Avg` column shows the average of the results of the previous columns. `none` indicates no corruption. We use the models trained under the `SGD/none` setting (first row) as baselines to calculate the $\mathrm{CE}_c^f$. The last five rows are the 5 best training corruptions ranked by the results in the `Avg` column.

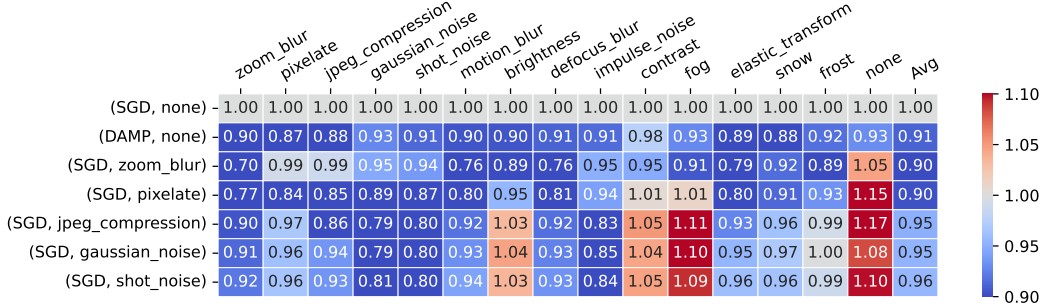

Figure 6: **DAMP improves robustness to all corruptions while preserving accuracy on clean images.** Results of PreActResNet18/TinyImageNet experiments averaged over 3 seeds. The heatmap shows $\mathrm{CE}_c^f$ described in Eq. (18), where each row corresponds to a tuple of training (`method`, `corruption`), while each column corresponds to the test corruption. The `Avg` column shows the average of the results of the previous columns. `none` indicates no corruption. We use the models trained under the `SGD/none` setting (first row) as baselines to calculate the $\mathrm{CE}_c^f$. The last five rows are the 5 best training corruptions ranked by the results in the `Avg` column.

**ImageNet-C (Hendrycks and Dietterich, 2019)** This dataset contains the corrupted versions of the ImageNet validation set, as the labels of the true ImageNet test set was never released. It contains 15 types of corruption, each divided into 5 levels of severity.

**Algorithm 3** DAAP: Data Augmentation via Additive Perturbations

1: **Input:** training data $\mathcal{S} = \{(\mathbf{x}_k, y_k)\}_{k=1}^{N}$, a neural network $\mathbf{f}(\cdot; \boldsymbol{\omega})$ parameterized by $\boldsymbol{\omega} \in \mathbb{R}^P$, number of iterations $T$, step sizes $\{\eta_t\}_{t=1}^{T}$, number of sub-batch $M$, batch size $B$ divisible by $M$, a noise distribution $\boldsymbol{\Xi} = \mathcal{N}(\mathbf{0}, \sigma^2 \mathbf{I}_P)$, weight decay coefficient $\lambda$, a loss function $\mathcal{L} : \mathbb{R}^P \to \mathbb{R}_+$.
2: **Output:** Optimized parameter $\boldsymbol{\omega}^{(T)}$.
3: Initialize parameter $\boldsymbol{\omega}^{(0)}$.
4: **for** $t = 1$ **to** $T$ **do**
5:     Draw a mini-batch $\mathcal{B} = \{(\mathbf{x}_b, y_b)\}_{b=1}^{B} \sim \mathcal{S}$.
6:     Divide the mini-batch into $M$ *disjoint sub-batches* $\{\mathcal{B}_m\}_{m=1}^{M}$ of equal size.
7:     **for** $m = 1$ **to** $M$ **in parallel do**
8:         Draw a noise sample $\boldsymbol{\xi}_m \sim \boldsymbol{\Xi}$.
9:         Compute the gradient $\mathbf{g}_m = \nabla_{\boldsymbol{\omega}} \mathcal{L}(\boldsymbol{\omega}; \mathcal{B}_m)\big|_{\boldsymbol{\omega}^{(t)} + \boldsymbol{\xi}}$.
10:     **end for**
11:     Compute the average gradient: $\mathbf{g} = \frac{1}{M} \sum_{m=1}^{M} \mathbf{g}_m$.
12:     Update the weights: $\boldsymbol{\omega}^{(t+1)} = \boldsymbol{\omega}^{(t)} - \eta_t \left(\mathbf{g} + \lambda \boldsymbol{\omega}^{(t)}\right)$.
13: **end for**

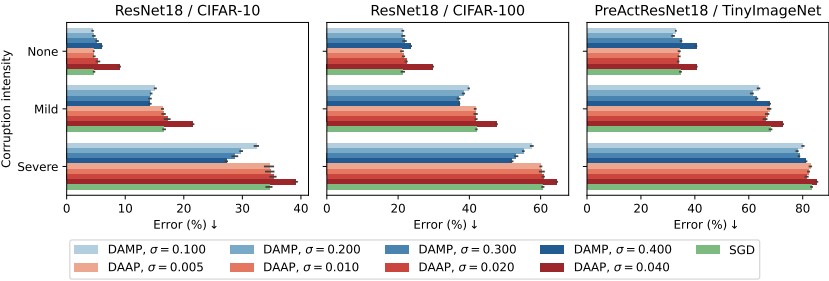

Figure 7: **DAMP has better corruption robustness than DAAP.** We report the predictive errors (lower is better) averaged over 5 seeds. None indicates no corruption. Mild includes severity levels 1, 2 and 3. Severe includes severity levels 4 and 5. We evaluate DAMP and DAAP under different noise standard deviations $\sigma$. These results imply that the multiplicative weight perturbations of DAMP are more effective than the additive perturbations of DAAP in improving robustness to corruptions.

**ImageNet-$\overline{\text{C}}$ (Mintun et al., 2021)** This dataset contains the corrupted versions of the ImageNet validation set, as the labels of the true ImageNet test set was never released. It contains 10 types of corruption, each divided into 5 levels of severity. The types of corruption in ImageNet-$\overline{\text{C}}$ differ from those in ImageNet-C.

**ImageNet-A (Hendrycks et al., 2021)** This dataset contains natural adversarial examples, which are real-world, unmodified, and naturally occurring examples that cause machine learning model performance to significantly degrade. The images contain in this dataset, while differ from those in the ImageNet validation set, stills belong to the same set of classes.

**ImageNet-D (Zhang et al., 2024)** This dataset contains images belong to the classes of ImageNet but they are modified by diffusion models to change the background, material, and texture.

**ImageNet-Cartoon and ImageNet-Drawing (Salvador and Oberman, 2022)** This dataset contains the drawing and cartoon versions of the images in the ImageNet validation set.

**ImageNet-Sketch (Wang et al., 2019)** This dataset contains sketch images belonging to the classes of the ImageNet dataset.

**ImageNet-Hard (Taesiri et al., 2023)** This dataset comprises an array of challenging images, curated from several validation datasets of ImageNet.

# F Training details

For each method and each setting, we tune the important hyperparameters ($\sigma$ for DAMP, $\rho$ for SAM and ASAM) using $10\%$ of the training set as validation set.

**CIFAR-10/100** For each setting, we train a ResNet18 for 300 epochs. We use a batch size of 128. We use a learning rate of $0.1$ and a weight decay coefficient of $5 \times 10^{-4}$. We use SGD with Nesterov momentum as the optimizer with a momentum coefficient of $0.9$. The learning rate is kept at $0.1$ until epoch 150, then is linearly annealed to $0.001$ from epoch 150 to epoch 270, then kept at $0.001$ for the rest of the training. We use basic data preprocessing, which includes channel-wise normalization, random cropping after padding and random horizontal flipping. On CIFAR-10, we set $\sigma = 0.2$ for DAMP, $\rho = 0.045$ for SAM and $\rho = 1.0$ for ASAM. On CIFAR-100, we set $\sigma = 0.1$ for DAMP, $\rho = 0.06$ for SAM and $\rho = 2.0$ for ASAM. Each method is trained on a single host with 8 Nvidia V100 GPUs where the data batch is evenly distributed among the GPUs at each iteration (data parallelism). This means we use the number of sub-batches $M = 8$ for DAMP.

**TinyImageNet** For each setting, we train a PreActResNet18 for 150 epochs. We use a batch size of 128. We use a learning rate of $0.1$ and a weight decay coefficient of $2.5 \times 10^{-4}$. We use SGD with Nesterov momentum as the optimizer with a momentum coefficient of $0.9$. The learning rate is kept at $0.1$ until epoch 75, then is linearly annealed to $0.001$ from epoch 75 to epoch 135, then kept at $0.001$ for the rest of the training. We use basic data preprocessing, which includes channel-wise normalization, random cropping after padding and random horizontal flipping. We set $\sigma = 0.2$ for DAMP, $\rho = 0.2$ for SAM and $\rho = 3.0$ for ASAM. Each method is trained on a single host with 8 Nvidia V100 GPUs where the data batch is evenly distributed among the GPUs at each iteration (data parallelism). This means we use the number of sub-batches $M = 8$ for DAMP.

**ResNet50 / ImageNet** We train each experiment for 90 epochs. We use a batch size of 2048. We use a weight decay coefficient of $1 \times 10^{-4}$. We use SGD with Nesterov momentum as the optimizer with a momentum coefficient of $0.9$. We use basic Inception-style data preprocessing, which includes random cropping, resizing to the resolution of $224 \times 224$, random horizontal flipping and channel-wise normalization. We increase the learning rate linearly from $8 \times 10^{-4}$ to $0.8$ for the first 5 epochs then decrease the learning rate from $0.8$ to $8 \times 10^{-4}$ using a cosine schedule for the remaining epochs. All experiments were run on a single host with 8 Nvidia V100 GPUs and we set $M = 8$ for DAMP. We use $p = 0.05$ for Dropout, $\sigma = 0.1$ for DAMP, $\rho = 0.05$ for SAM, and $\rho = 1.5$ for ASAM. We also use the image resolution of $224 \times 224$ during evaluation.

**ViT-S16 / ImageNet / Basic augmentations** We follow the training setup of Beyer et al. (2022) with one difference is that we only use basic Inception-style data processing similar to the ResNet50/ImageNet experiments. We use AdamW as the optimizer with $\beta_1 = 0.9$, $\beta_2 = 0.999$ and $\epsilon = 10^{-8}$. We clip the gradient norm to $1.0$. We use a weight decay coefficient of $0.1$. We use a batch size of 1024. We increase the learning rate linearly from $10^{-6}$ to $10^{-3}$ for the first 10000 iterations, then we anneal the learning rate from $10^{-3}$ to $0$ using a cosine schedule for the remaining iterations. We use the image resolution of $224 \times 224$ for both training and testing. Following Beyer et al. (2022), we make 2 minor modifications to the original ViT-S16 architecture: (1) We change the position embedding layer from `learnable` to `sincos2d`; (2) We change the input of the final classification layer from the embedding of the `[cls]` token to global average-pooling. All experiments were run on a single host with 8 Nvidia V100 GPUs and we set $M = 8$ for DAMP. We use $p = 0.10$ for Dropout, $\sigma = 0.25$ for DAMP, $\rho = 0.6$ for SAM, and $\rho = 3.0$ for ASAM.

**ViT-S16 and B16 / ImageNet / MixUp and RandAugment** Most of the hyperparameters are identical to the ViT-S16 / ImageNet / Basic augmentations setting. With ViT-S16, we use $p = 0.1$ for Dropout, $\sigma = 0.10$ for DAMP, $\rho = 0.015$ for SAM, and $\rho = 0.4$ for ASAM. With ViT-B16, we use $p = 0.1$ for Dropout, $\sigma = 0.15$ for DAMP, $\rho = 0.025$ for SAM, and $\rho = 0.6$ for ASAM.

