# OpenReview forum: "Improving robustness to corruptions with multiplicative weight perturbations"
_NeurIPS.cc/2024/Conference — NeurIPS 2024 spotlight_

### Official Review · Reviewer_dwU7 · 2024-07-01

**Soundness:** 3
**Presentation:** 4
**Contribution:** 3
**Rating:** 6
**Confidence:** 4

**Summary:**

To achieve robustness against corruptions, most of existing works relying on incorporating some corruptions during the training. In contrast, this paper takes a different perspective and propose a weight-based approach called DAMP to achieve model robustness against corruptions without compromising accuracy on clean examples.

This work also studies Sharpness-Aware Minimisation (SAM) methods and point out the relationship between their proposed DAMP and SAM-based methods.

**Strengths:**

The paper takes a new perspective compared to existing methods to achieve robustness against corruptions.

The proposed method is simple with nearly similar complexity to SGD and achieve competitive results.

Theoretical results is provided to justify the hypothesis.

The presentation is clear and easy to follow.

**Weaknesses:**

There could be a related work [1]  focusing on (adversarial) corruptions, where it also approaches the model robustness problem from model weights perspective. I suggest the authors to take a look at [1] and compare with their work.

The experimental results relying on only algorithmically corruption (ImageNet-C, CIFAR-C) could be limited. I suggest the author to conduct the results on other corruption (such as Stylised ImageNet, ImageNet-D), natural corruption (such as ImageNet-A, ImageNet-Sketch), and adversarial corruption. I believe this assessment will further improve the experimental results.

The results for SAM/ASAM in Tables 1 and 2 are relatively comparable to those for DAMP when considering the effect of accuracy on clean images. As we know, improving accuracy on clean images generally leads to better accuracy on corrupted images. While I understand that this work takes a different approach from SAM/ASAM, I am curious to know what advantages DAMP offers over SAM/ASAM.

As the theory based on a simple feedforward neural network, I wonder if the DAMP can improve the robustness on deeper/modern neural networks, such as Resnet-152, EfficientNet, ConvNext, MaxViT?

[1] Ha, Hyeonjeong, Minseon Kim, and Sung Ju Hwang. "Generalizable lightweight proxy for robust NAS against diverse perturbations." Advances in Neural Information Processing Systems 36 (2024).

**Questions:**

What is the effectiveness of DAMP on deeper/modern neural networks, such as Resnet-152 EfficientNet, ConvNext, MaxViT?

What is the effectiveness of DAMP on other corruption (such as Stylised ImageNet, ImageNet-D), natural corruption (such as ImageNet-A, ImageNet-Sketch), and adversarial corruption?

What is the advantage of DAMP over SAM/ASAM?

---

> ### Author Rebuttal · Authors · 2024-08-06
>
> **1. "There could be a related work [1] focusing on (adversarial) corruptions, where it also approaches the model robustness problem from model weights perspective...."**
>
> Thank you for the suggestion. This is indeed an interesting related paper which proposes a score called CRoZe that can quickly evaluate the robustness to input perturbations of an architecture using only one batch of training data for zero-shot neural architecture search (NAS). We think that the multiplicative weight perturbations (MWPs) of DAMP can also be used inside the calculation of the CRoZe score to evaluate the robustness of an architecture, which could be a good future research direction. We will include this mention of [1] in the Related Works section of our revised paper.
>
> **2. "What is the effectiveness of DAMP on deeper/modern neural networks, such as Resnet-152 EfficientNet, ConvNext, MaxViT?"**
>
> In our paper, we have demonstrated the effectiveness of DAMP on ResNet50 and ViT-S16 on ImageNet. We now further provide new results with ViT-B16 in our global response (Table D in the PDF). These results together demonstrate that DAMP is capable of improving the robustness of deep and modern neural networks. Furthermore, as modern networks get larger and have bigger learning capacity, we believe that they could benefit from the additional implicit data augmentations/perturbations induced by the random multiplicative weight perturbations of DAMP, even more so than their smaller counterparts since they have the capacity to learn to output the correct predictions even when the inputs are corrupted by these implicit perturbations.
>
> **3. "What is the effectiveness of DAMP on other corruption (such as Stylised ImageNet, ImageNet-D), natural corruption (such as ImageNet-A, ImageNet-Sketch), and adversarial corruption?"**
>
> Thank you for your suggestions. We provide new results in our global response (Table B and D in the PDF) where we evaluate models trained with DAMP and baseline methods on ImageNet-D [2], ImageNet-A [3], ImageNet-Sketch [4], ImageNet-Cartoon [5], ImageNet-Drawing [5], and adversarial corruptions generated by FGSM [6], demonstrating that models trained with DAMP are more robust to various types of input perturbations than the baseline training methods.
>
> **4. "What is the advantage of DAMP over SAM/ASAM?"**
>
> As stated in our paper, the main advantage of DAMP over SAM/ASAM is that it requires only one forward-backward pass per training iteration, while SAM/ASAM requires two consecutive forward-backward passes per iteration. Therefore, given the same number of iterations, SAM/ASAM takes twice as long to finish training than DAMP. Furthermore, our experimental results show that on corrupted inputs, DAMP outperforms SAM in the majority of cases and is competitive with ASAM. Finally, our new results in the global response (Table D in the attached PDF) indicate that training a ViT-B16 (86M params) with DAMP leads to better accuracy than training ViT-S16 (22M params) with SAM/ASAM, yet both settings take roughly the same amount of time. Thus given the same training budget, it is better to train a large model with DAMP than to train a smaller model with SAM/ASAM.
>
> [1] Ha et al. "Generalizable lightweight proxy for robust NAS against diverse perturbations." NeurIPS 2024.
>
> [2] Zhang et al. "ImageNet-D: Benchmarking Neural Network Robustness on Diffusion Synthetic Object". CVPR 2024.
>
> [3] Hendrycks et al. "Natural Adversarial Examples". CVPR 2021.
>
> [4] Wang et al. "Learning Robust Global Representations by Penalizing Local Predictive Power". NeurIPS 2019.
>
> [5] Salvador et al. "ImageNet-Cartoon and ImageNet-Drawing: two domain shift datasets for ImageNet". ICML 2022 Shift Happens Workshop.
>
> [6] Goodfellow et al. "Explaining and Harnessing Adversarial Examples". arXiv preprint arXiv:1412.6572, 2014.

---

> > ### Comment · Reviewer_dwU7 · 2024-08-09
> > **Official Comment by Reviewer dwU7**
> >
> > I thank the authors' effort in conducting new experiments, which have made the proposed method more convincing to me.
> >
> > However, the performance of DAMP does not appear to significantly surpass that of ASAM/SAM. I agree with the authors that DAMP generally offers a x2 speedup in iteration training time compared to ASAM/SAM. With the same training iterations (i.e., faster training time), ASAM/SAM still outperforms DAMP. DAMP only (marginally) surpasses ASAM/SAM when it is trained with more iterations, resulting in the same training time.
> >
> > Consequently, I appreciate the authors for their thorough rebuttal and have raised my score to 6.

---

> ### Author Response · Authors · 2024-08-09
>
> Thank you for the comments and for the increased confidence in the proposed method.
>
> We note that there are cases where DAMP does not need more training iterations to surpass ASAM/SAM. For instance, in the ResNet50 / ImageNet experiments (Table 1 in the paper), we use 90 epochs for all training methods, yet DAMP is able to perform competitively with ASAM and surpass SAM.
>
> Furthermore, we show in Table D of our global response that training ViT-S16 / ImageNet using 500 epochs of DAMP (which took 111hrs) outperforms 300 epochs of ASAM/SAM (which took 123hrs). Therefore, even in the cases where DAMP requires more iterations to outperform ASAM/SAM, it still ends up requiring less overall runtime than ASAM/SAM.

---

### Official Review · Reviewer_Vu1M · 2024-07-09

**Soundness:** 3
**Presentation:** 3
**Contribution:** 3
**Rating:** 6
**Confidence:** 3

**Summary:**

This submission tackles the problem of generalization of image classifiers to corrupted images. The paper shows a link between data augmentation as well as previous methods such as Adaptive Sharpness-aware Minimization (ASAM) and multiplicative weight perturbations. Effectively, ASAM works as an adversarial weight multiplication which requires a second gradient pass per batch. The proposed DAMP method relaxes this constraint by using random weight perturbations. DAMP is benchmarked on CIFAR-10/100, TinyImageNet, and ImageNet under common corruptions (and follow-up works) with ResNet and ViT where it often matches or even exceeds the original SAM at lower cost.

**Strengths:**

- Good writing.
- A theoretical motivation for DAMP is provided (arguably with under-simplified relaxation).
- A connection between ASAM and multiplicative weight perturbations is shown.
- Good evaluation: DAMP is benchmarked on CIFAR-10/100, TinyImageNet, and ImageNet under common corruptions (and follow-up works) with ResNet and ViT. It is also compared to SAM and ASAM. All experiments (except ViT) use multiple runs!
- DAMP outperforms SAM in most cases at a lower cost; it allows to training of ViTs without excessive augmentation techniques.
- DAMP is theoretically domain-agnostic and could be applied beyond image classification or vision.

**Weaknesses:**

My only concern about this work is the contextualization of this work. It seems like DAMP is mostly an approximation of ASAM which allows faster training (and that is great) but at the same time, it is not as effective as ASAM. So effectively, this method introduces a trade-off between performance (accuracy) and training time. This raises the following detailed concerns:

- If multiplicative weight perturbations are connected to data augmentation then how does DAMP rank against them? Augmentations are mostly cheap so they could achieve even better performance at the same overhead cost. One indication of this is in the ViT experiments on ImageNet: DAMP achieves a 23.7% clean error in 200 epochs without augmentations. This may sound impressive at first, however, the baseline that the authors rely on (Beyer, 2022) even exceeds this (23.5% error) in just 90 epochs. The only difference is that (Beyer, 2022) utilizes some simple augmentations. That is only clean accuracy but usually, it is indicative of performance under distribution shift [1]. Ideally, the authors would demonstrate that on a train time vs. mCE or clean acc curve, DAMP exceeds the current Pareto frontier (by benchmarking a few modern augmentation techniques in addition to SAM/ASAM).
- Additionally, it would be great to understand if DAMP is compatible with augmentations. I don't want to use the clearly stated Limitations against the authors but this is a very straightforward question that I had during reading.


Other than that the theory in Eq.1ff assumes bias-free MLP architectures without non-linearities which obviously does not scale to modern architectures. (The authors mention this in their limitations and this is not something that needs to be addressed, but it is a weakness nonetheless)

Nitpicks:
- L118: I agree that random perturbations introduce almost no overhead, but you shouldn't state "we found that ... had similar training times" in a scientific paper without backing it up with hard numbers.
- It would be great to add averages over all severities in Tab. 1 and 2.

Suggestion:
- A simple way to kill all my concerns is to show performance on non-vision tasks (since DAMP is in theory data-agnostic). In that case, it would exceed the value of domain-specific augmentations. This would greatly enhance this paper but of course, I do not expect this from a rebuttal.

[1] Taori et al., "Measuring Robustness to Natural Distribution Shifts in Image Classification", NeurIPS 2020.

**Questions:**

See above.

**Limitations:**

The authors are very upfront about Limitations and I have no other suggestions.

---

> ### Author Rebuttal · Authors · 2024-08-06
>
> **1. "...It seems like DAMP is mostly an approximation of ASAM which allows faster training (and that is great) but at the same time, it is not as effective as ASAM. So effectively, this method introduces a trade-off between performance (accuracy) and training time..."**
>
> There is a misunderstanding here. DAMP is actually **not an approximation of ASAM** and can even outperform ASAM. The objective function of DAMP (Eq. 16 in our paper) is:
> $$ \mathcal{L}\_{\mathrm{DAMP}}(\boldsymbol{\omega}; \mathcal{D}) = \mathbb{E}\_{\boldsymbol{\xi}\sim\mathcal{N}(\boldsymbol{1},\sigma^2\mathbf{I})}\left[\mathcal{L}(\boldsymbol{\omega}\circ\boldsymbol{\xi}; \mathcal{D})\right] + \frac{\lambda}{2}||\boldsymbol{\omega}||\_2^2$$
> while the objective function of ASAM (Eq. 21 in our paper) can be written as:
> $$\mathcal{L}\_{\mathrm{ASAM}}(\boldsymbol{\omega};\mathcal{D})=\max\_{||\boldsymbol{\xi}||\_2 \leq \rho}\mathcal{L}(\boldsymbol{\omega}\circ(1+\boldsymbol{\xi});\mathcal{D})+\frac{\lambda}{2}||\boldsymbol{\omega}||\_2^2$$
> where $\boldsymbol{\omega}$ is the model's parameters, $\mathcal{D}$ is the training data, $\mathcal{L}$ is the loss function such as cross-entropy, and $\circ$ denotes the Hadamard product. Thus DAMP minimizes the **expected loss** under multiplicative weight perturbations (MWPs), while ASAM minimizes the loss under the **worse-case MWP**. This is a crucial distinction, since minimizing the expectation allows us to devise an efficient version of DAMP (Algorithm 1 in the paper), while ASAM needs two forward-backward passes for each iteration (the first one to approximate the worse-case MWP, the second one is to minimize the loss under such MWP). Thus given the same number of training iterations, ASAM takes **twice as long** to train as DAMP. In Table D of the PDF attached to our global response, training a ViT-B16 (86M params) with DAMP leads to better accuracy than training ViT-S16 (22M params) with ASAM, yet both settings take roughly the same amount of time. Thus given the same training budget, it is better to train a large model with DAMP than to train a smaller model with ASAM.
>
> **2. "If multiplicative weight perturbations are connected to data augmentation then how does DAMP rank against them?..."**
>
> **"Additionally, it would be great to understand if DAMP is compatible with augmentations..."**
>
> Modern data augmentations (DAs) (such as MixUp, RandAugment) are cheap and contain informative prior knowledge, and thus they could greatly improve performance of a model. However, they are specific to computer vision and not applicable to other domains like natural language processing. On the other hand, multiplicative weight perturbations (MWPs) are less informative than DAs but they are domain-agnostic and thus can be applied to a wide range of tasks. For instance, DAMP improves the test accuracy of Mistral-7B finetuned by LoRA on the MedMCQA dataset from 49.05% to 50.25%, as shown in Table A in the PDF attached to the global response.  Furthermore, since MWPs operate on the weight space while DAs operate on the input space, they can be combined together to further enhance performance, as shown in Table D of the global response combining DAMP with MixUp and RandAugment to train ViT-S16 and ViT-B16.
>
> **3.  "the theory in Eq.1ff assumes bias-free MLP architectures without non-linearities which obviously does not scale to modern architectures."**
>
> There is a misunderstanding here. In our theoretical analysis, we use a bias-free MLP architectures with **non-linear activations**, as shown in Eq. 1-3 in our paper. It would be pointless for us to analyze an MLP without non-linear activations since such model is equivalent to a simple one-layer linear model. We note that other papers (e.g. [1, 2]) also assume a bias-free MLP for their analyses.  Finally, we want to note that Theorem 1 in our paper motivates the design of the DAMP algorithm which, as we demonstrate through our large-scale ImageNet experiments, does scale well to modern architectures.
>
> **4. "I agree that random perturbations introduce almost no overhead, but you shouldn't state "we found that ... had similar training times" in a scientific paper without backing it up with hard numbers."**
>
> **"It would be great to add averages over all severities in Tab. 1 and 2."**
>
> Thank you for your suggestions. We now added the runtimes and the results averaged over severities for each setting in our global response (Table B and D), and will include them in the revised paper.
>
> **5. "A simple way to kill all my concerns is to show performance on non-vision tasks (since DAMP is in theory data-agnostic). In that case, it would exceed the value of domain-specific augmentations. This would greatly enhance this paper but of course, I do not expect this from a rebuttal."**
>
> Thank you for your suggestions. We now added results of using DAMP with LoRA to finetune a Mistral-7B on the MedMCQA dataset in our global response (Table A), demonstrating that it improves the test accuracy from 49.05% to 50.25%.
>
> [1] Dusenberry et al. "Efficient and scalable Bayesian neural nets with rank-1 factors." ICML, 2020.
>
> [2] Andriushchenko et al. "Sharpness-Aware Minimization Leads to Low-Rank Features". NeurIPS, 2023.

---

> > ### Comment · Reviewer_Vu1M · 2024-08-08
> >
> > I thank the authors for their rebuttal. The newly added experiments address my concerns - I'd suggest including them in the paper. I am raising my score to 6.

---

> ### Author Response · Authors · 2024-08-08
>
> Thank you for the comments. We will definitely include these new results in the revised version of the paper.

---

### Official Review · Reviewer_dM3x · 2024-07-13

**Soundness:** 3
**Presentation:** 3
**Contribution:** 3
**Rating:** 6
**Confidence:** 2

**Summary:**

The paper presents Data Augmentation via Multiplicative Perturbations (DAMP), a novel method to enhance DNN robustness against image corruptions by optimizing with random multiplicative weight perturbations. This approach improves generalization on corrupted images without compromising accuracy on clean ones. The authors show that input perturbations can be mimicked by multiplicative perturbations in the weight space. The authors demonstrate DAMP's effectiveness across multiple datasets and architectures, and explore its connection to Adaptive Sharpness-Aware Minimization (ASAM).

**Strengths:**

The paper presents a novel perspective on enhancing neural network robustness by leveraging multiplicative weight perturbations. The experiments are comprehensive and well-designed, covering multiple datasets and model architectures. The results consistently show that DAMP improves robustness against a wide range of corruptions without sacrificing performance on clean images. The method achieves these improvements without incurring additional computational costs.

**Weaknesses:**

It would be beneficial to see how the method performs with different hyperparameter values, as the reported numbers for different metrics are close to each other. Assumption 2 is not explained very well and could benefit from a clearer, more detailed explanation. While DAMP works well for small models and datasets, it would be interesting to see the results with larger models and datasets, as it seems to show some instability in results on these larger models and datasets.

**Questions:**

Why is DAMP trained for 200 epochs compared to other methods trained for 100 epochs in Table 2?
How does the model perform when tested on other types of distribution shifts beyond those included in the corruption benchmarks?
How do the methods perform without basic Inception-style preprocessing?

**Limitations:**

The results could be better explained, particularly why the accuracy on clean images and lower severity corruptions is lower than the ASAM method in larger-scale experiments for ResNet/ImageNet.

---

> ### Author Rebuttal · Authors · 2024-08-06
>
> **1. "It would be beneficial to see how the method performs with different hyperparameter values, as the reported numbers for different metrics are close to each other."**
>
> Thank you for your suggestion. We provide Table C in the PDF file attached to our global response showing the accuracy of ViT-S16 on ImageNet under different variances of the random multiplicative weight perturbations of DAMP. Specifically, as we increase the perturbation variance, the accuracy improves up to a maximum value then degrades afterwards. We will include this table in the revised paper.
>
> **2.  "Assumption 2 is not explained very well and could benefit from a clearer, more detailed explanation"**
>
> Assumption 2 states that the gradient with respect to the input $\mathbf{x}$ of the loss function $\ell(\boldsymbol{\omega}, \mathbf{x}, y)$ is Lipschitz continuous. This simply means that there exists a constant $C > 0$ such that for all $\mathbf{x}\_1, \mathbf{x}\_2$, we have:
> $$||\nabla\_{\mathbf{x}}\ell(\boldsymbol{\omega},\mathbf{x}\_1,y)-\nabla\_{\mathbf{x}}\ell(\boldsymbol{\omega},\mathbf{x}\_2,y)||\_2^2\leq C||\mathbf{x}\_1-\mathbf{x}\_2||\_2^2$$
> We will clarify assumption 2 as shown here in the revised paper.
>
> **3. "While DAMP works well for small models and datasets, it would be interesting to see the results with larger models and datasets, as it seems to show some instability in results on these larger models and datasets."**
>
> Perhaps there is a misunderstanding here. We have shown that DAMP works well on both ResNet50 and ViT-S16 on ImageNet in our paper, and we experienced no instability when using DAMP to train these models. In our new experiments (see Table D in the PDF file of our global response), we use DAMP to train a ViT-B16, a larger version of ViT-S16, showing that DAMP also leads to improved performance in this case. Thus we believe that DAMP is perfectly capable of training large models on large datasets.
>
> **4. "Why is DAMP trained for 200 epochs compared to other methods trained for 100 epochs in Table 2?"**
>
> In Table 2, we compare models trained with DAMP and AdamW for 200 epochs to models trained with SAM and ASAM for 100 epochs. This is because SAM and ASAM require two forward-backward passes per training iteration, while DAMP and AdamW require only one forward-backward pass per iteration. Therefore, the time it takes to run 200 epochs of DAMP is roughly the same as 100 epochs of SAM/ASAM. We have already emphasized this in the paper but will add a further comment. We also have included the runtime of these experiments in Table B and Table D of the PDF attached to our global response.
>
> **5. "How does the model perform when tested on other types of distribution shifts beyond those included in the corruption benchmarks?"**
>
> We provide new results in Table B and Table D in the PDF of our global response where we evaluate models trained with DAMP and baseline methods on ImageNet-D [1], ImageNet-A [2], ImageNet-Sketch [3], ImageNet-Cartoon [4], ImageNet-Drawing [4], and adversarial corruptions generated by FGSM [5], demonstrating that models trained with DAMP are more robust to various types of input perturbations than the baseline training methods.
>
> **6. "How do the methods perform without basic Inception-style preprocessing?"**
>
> We provide new results in Table D in the PDF of our global response where we train ViT-S16 and ViT-B16 with MixUp and RandAugment, showing that DAMP is able to work in tandem with these modern data augmentations to further enhance model performance.
>
> **7. "The results could be better explained, particularly why the accuracy on clean images and lower severity corruptions is lower than the ASAM method in larger-scale experiments for ResNet/ImageNet."**
>
> As we explain in Section 3 of the paper, ASAM actually minimizes the training loss under adversarial multiplicative weight perturbations (MWPs), as shown by the inner maximization over the MWPs $\boldsymbol{\xi}$ in its objective function (Eq. 21 in the paper):
> $$\mathcal{L}\_{\mathrm{ASAM}}(\boldsymbol{\omega};\mathcal{D})=\max\_{||\boldsymbol{\xi}||\_2 \leq \rho}\mathcal{L}(\boldsymbol{\omega}\circ(1+\boldsymbol{\xi});\mathcal{D})+\frac{\lambda}{2}||\boldsymbol{\omega}||\_2^2$$
> and thus it is also able to produce robust models. This is why Table 1 in the paper (ResNet50 / ImageNet) shows that ASAM is able to outperform DAMP on clean images and on corruption severity levels 1, 2 and 3 of ImageNet-C. However, Table 1 also shows that DAMP outperforms ASAM on severity levels  4 and 5 of ImageNet-C as well as on all 5 severity levels of ImageNet-$\bar{\mathrm{C}}$. Furthermore, as we have stated above, the training time of DAMP is half that of ASAM, and since we use 90 epochs for all experiments in Table 1, this means that DAMP is able to outperform ASAM on the majority of test sets while being more efficient. We will add a further pointer to these for clarity.
>
> [1] Zhang et al. "ImageNet-D: Benchmarking Neural Network Robustness on Diffusion Synthetic Object". CVPR 2024.
>
> [2] Hendrycks et al. "Natural Adversarial Examples". CVPR 2021.
>
> [3] Wang et al. "Learning Robust Global Representations by Penalizing Local Predictive Power". NeurIPS 2019.
>
> [4] Salvador et al. "ImageNet-Cartoon and ImageNet-Drawing: two domain shift datasets for ImageNet". ICML 2022 Shift Happens Workshop.
>
> [5] Goodfellow et al. "Explaining and Harnessing Adversarial Examples". arXiv preprint arXiv:1412.6572, 2014.

---

> > ### Comment · Reviewer_dM3x · 2024-08-12
> >
> > Thank you for the clarifications provided in your rebuttal. After reviewing the rebuttal and considering other comments, I will maintain my current rating. Including SAM/ASAM+ViT-B16 in Table D would further improve the paper.

---

> > > ### Author Response · Authors · 2024-08-12
> > >
> > > Thank you for your comments. We will definitely include results of SAM/ASAM+ViT-B16 in the revised version of the paper.

---

### Official Review · Reviewer_XEeL · 2024-07-13

**Soundness:** 3
**Presentation:** 2
**Contribution:** 2
**Rating:** 5
**Confidence:** 4

**Summary:**

This paper works on improving robustness by multiplying a random Gaussian variable on weights during training.

**Strengths:**

The writing is easy to read and follow.

**Weaknesses:**

1. The novelty is quite limited. This concept has been proposed and explored for at least a decade. For instance, variational dropout already suggested this approach, using a Bernoulli distribution. Another example can be found in [1], "Multiplicative or Additive Perturbation?", where they used a Gaussian distribution.

2. The theoretical foundation is weak. The theory section consists of trivial analysis by adding constrained corruption to inputs, which is not significantly related to this work.

Reference:

1. Dusenberry, Michael, et al. "Efficient and scalable Bayesian neural nets with rank-1 factors." International Conference on Machine Learning. PMLR, 2020.

**Questions:**

1. What is the novelty of this work?

2. How does DAMP compare to modern augmentation techniques such as those used in DeiT and subsequent works? DAMP's augmentation does not appear to be simpler than these methods. Furthermore, DAMP is evidently more unstable, particularly in large-scale training, due to the stochasticity introduced during training, a common issue observed in many Bayesian Neural Network (BNN) related works.

**Limitations:**

See the above sections.

---

> ### Author Rebuttal · Authors · 2024-08-06
>
> **1. On the novelty of this work and its connections to previous works:**
>
> The novel contributions are:
> 1. Showing the theoretical connection between perturbations in the input space and perturbations in the weight space, in particular, that one can simulate perturbations in the input space via multiplicative weight perturbations.
> 2. From this connection, we propose a new method, Data Augmentation via Multiplicative Perturbations (DAMP), which multiplies each individual weight of a network with its own Gaussian random variables during training.
> 3. We provide a new theoretical interpretation of Adaptive Sharpness Aware Minimization (ASAM) [3], a variant of SAM [2], that ASAM actually minimizes the loss under adversarial *multiplicative* weight perturbation as opposed to SAM which minimizes the loss under adversarial *additive* perturbations.
> 4. We show that DAMP works well with modern architectures as demonstrated by our large-scale experiments on training ResNet50 and ViT-S16 on ImageNet.
>
> The differences between DAMP and previous works like Dropout and its variants are:
> 1. Dropout multiplies *all the weights connecting to the same input node* with a *random Bernoulli variable*, which has the disadvantage that it reduces the update frequency of the weights during training (since dropped weights receive no gradient update from backpropagation). On the other hand, DAMP multiplies *each individual weight* with *its own random Gaussian variable*, which perturbs the weights while allowing them to receive non-zero gradients at each iteration.
> 2. Dropout focuses on improving generalization on clean test data, while DAMP focuses on improving generalization on corrupted test data.
> 3. The work in [1] multiplies *all the weights connecting to the same input node* with a *Gaussian random variable*. Their motivation is to introduce Rank-1 BNNs, an efficient Bayesian neural network (BNN), and they attribute the improvements on corrupted data to the epistemic uncertainty of the Rank-1 BNNs. Our work demonstrates that *multiplicative weight perturbations* are actually the main reason for the improved robustness.
>
> While the concept of weight perturbations is not new, to the best of our knowledge, no earlier work specifically studies this technique to robustify models to input corruptions. Here we shed light on a simple algorithm that could be applied to any training setting to enhance model robustness with zero additional cost, and we believe that this new knowledge is beneficial to the machine learning community.
>
> **2. "The theoretical foundation is weak. The theory section consists of trivial analysis by adding constrained corruption to inputs, which is not significantly related to this work."**
>
> We appreciate your perspective on our theoretical analysis. However, we believe that our analysis, while simple, provides a direct link between input corruptions and multiplicative weight perturbations. Specifically, Theorem 1 proves that the training loss under input corruptions is upper bounded by the training loss under multiplicative weight perturbations plus an $L2$-regularization term. This connection serves as the foundation for our DAMP algorithm, offering a clear rationale for its design and effectiveness. We also note that many analyses, when presented in a clear and concise manner, may seem obvious in hindsight.
>
> **3. Comparing DAMP to modern augmentation techniques:**
>
> The modern augmentation techniques directly modify the inputs in the input space, while DAMP perturbs the weights in the weight space to simulate input corruptions. As they operate on two different spaces, DAMP and the augmentation techniques can be combined together to further enhance corruption robustness. We demonstrate this idea with new experiments training the ViT-S16 and ViT-B16 on ImageNet with MixUp and RandAugment (Table D in the PDF of our global response), showing that DAMP works in tandem with these augmentations to boost robustness.
>
> DAMP is simpler than these augmentation techniques since it is domain-agnostic and can be applied to any tasks and any neural network architectures. This is actually an advantage as demonstrated through our new experiments finetuning Mistral-7B on MedMCQA dataset using DAMP and LoRA (Table A in the PDF of our global response). By applying DAMP on the low-rank weights, we improve the test accuracy on the MedMCQA from 49.05% to 50.25%.
>
> **4. "DAMP is evidently more unstable, particularly in large-scale training, due to the stochasticity introduced during training..."**
>
> We respectfully disagree. DAMP multiplies each individual weight with its own Gaussian random variable with distribution $\mathcal{N}(1, \sigma^2)$, and thus we can control the strength of the perturbations via the variance $\sigma^2$. Of course, if we set $\sigma^2$ too high then the training cannot converge due to high stochasticity, while setting $\sigma^2$ too low reduces the effectiveness of the perturbations. In this sense, our method is no different than other regularization techniques, as setting the dropout rate or the weight decay coefficient too high also prevents convergence. In our experiments, we never observe any instability when the variance is properly tuned using a validation set, and we were able to successfully train ResNet50, ViT-S16, and ViT-B16 on ImageNet, as well as finetuning Mistral-7B. Finally, we note that earlier studies such as [4] showed that introducing stochasticity in neural network training acts as regularization which improves generalization.
>
> [1] Dusenberry et al. "Efficient and scalable Bayesian neural nets with rank-1 factors." ICML, 2020.
>
> [2] Foret et al. “Sharpness-Aware Minimization for Efficiently Improving Generalization”. ICLR, 2021
>
> [3] Kwon et al. “ASAM: Adaptive Sharpness-Aware Minimization for Scale-Invariant Learning of Deep Neural Networks”. ICML, 2022
>
> [4] Keskar et al. “On Large-Batch Training for Deep Learning: Generalization Gap and Sharp Minima”, ICLR 2017.

---

> > ### Comment · Reviewer_XEeL · 2024-08-13
> >
> > Thank you to the authors for their response. Unfortunately, it does not address my concerns regarding this paper.
> >
> > Is the technique new? No. This is simply variational dropout (VD). While the authors cite the paper by Kingma et al. (2015) in the related work section, their comments are somewhat misleading. The VD series of works has already established that traditional dropout involves multiplying Bernoulli variables, and that these can also be Gaussian variables, which is exactly what this work does. However, the authors only cite Kingma’s paper but overlook this point, merely mentioning that dropout involves multiplying a Bernoulli variable.
> >
> > It's well-established that this is just a variant of dropout. Is using dropout to improve robustness new? Certainly not. A simple search for "dropout improve robustness" yields numerous papers and blog posts on this topic. Even for variational dropout, this approach isn't novel. Variational dropout is a standard technique in training Bayesian neural networks (BNNs), where testing robustness is a common metric, as BNNs are well-known for their enhanced robustness.
> >
> > Is a weight-based approach to improve robustness new? No. Dropout itself can be considered a weight-based approach to improving robustness, and it is standard practice to combine dropout with other techniques applied to inputs.
> >
> > Just using Mixup and Randaugment seems not to be a strong baseline. What about stronger baseline such as the commonly-used DeiT [1]? Actually they mentioned that dropout hurts performance in their settings.
> >
> > 1. Training data-efficient image transformers & distillation through attention

---

> > > ### Author Response · Authors · 2024-08-13
> > >
> > > Thank you for your comments.
> > >
> > > **1. On the differences between DAMP, Variational Dropout (VD) and Dropout**
> > >
> > > We need to emphasize that there is an important difference between DAMP, VD and Dropout. More details below:
> > >
> > > While it is true that DAMP is closely related to the family of Dropout methods (which includes VD) as we have stated in the Related works section of our paper, we also highlight one crucial difference: DAMP **multiplies each individual weight with its own Gaussian random variable (RV)** $\mathcal{N}(1, \sigma^2)$, while VD and Dropout **share the same multiplicative RV for all weights connecting to the same input node** (VD uses Gaussian RVs while Dropout uses Bernoulli RVs). This difference is very important because we have shown that DAMP **can train a ViT-S16 from scratch on ImageNet to match performance of a ResNet50 without strong data augmentions**, while Dropout is unable to do so as stated in Section 3 of [1]  which we directly quote below:
> > >
> > > >  Existing works report that ViTs yield inferior accuracy to the ConvNets of similar size and throughput when trained from scratch on ImageNet without the combination of those advanced data augmentations, despite using various regularization techniques (e.g., large weight decay, Dropout (Srivastava et al., 2014), etc.).
> > >
> > > This phenomenon suggests that the weight perturbations of DAMP are more expressive than VD and Dropout.
> > >
> > > The reason we did not mention VD further in our rebuttal is because we already mentioned [2] which is a more recent work that uses Gaussian multiplicative weight perturbations. We also emphasize that this work also **shares the same multiplicative Gaussian RVs for all weights connecting to the same input node** just like VD and therefore is different from DAMP.
> > >
> > > Regarding Bayesian neural networks (BNNs), which you mention as the reason why VD improves robustness, they achieve robustness by ensembling multiple predictions from samples drawn from their weight posteriors via Bayesian model averaging [3]. Thus at test time, BNNs need to make multiple forward passes to make a prediction on each test input. Our method directly improves the robustness of deterministic neural networks, which only need one forward pass to make a prediction on each test input.
> > >
> > > [1] Chen et al. "When Vision Transformers Outperform ResNets without Pre-training or Strong Data Augmentations". ICLR 2022.
> > >
> > > [2] Dusenberry et al. "Efficient and scalable Bayesian neural nets with rank-1 factors." ICML, 2020.
> > >
> > > [3] Wilson et al. “Bayesian Deep Learning and a Probabilistic Perspective of Generalization.” 	arXiv:2002.08791. 2022
> > >
> > > **2. On the novelty of our paper**
> > >
> > > In addition to introducing a new training method (DAMP), the novelty of our work also includes providing a new perspective on Adaptive Sharpness Aware Minimization (ASAM) [4], showing that it optimizes neural networks under adversarial **multiplicative** weight perturbations, while its predecessor Sharpness Aware Minimization (SAM) [5] optimizes under adversarial **additive** weight perturbations. This explains why both DAMP and ASAM outperform SAM on corrupted test sets as we showed in our paper.
> > >
> > > Furthermore, while we agree that Dropout-like weight-based perturbation approach to robustness is not new, most prior works we could find only consider Bernoulli random variables. This is also evidenced by the fact that the training recipes of large vision models such as ViT only consider using Dropout and not the alternatives. Here we shed light on DAMP which is possibly a better alternative of Dropout and we demonstrate that DAMP greatly improves performance and robustness of ViT, regardless of whether strong data augmentation techniques are applied.
> > >
> > > [4] Kwon et al. “ASAM: Adaptive Sharpness-Aware Minimization for Scale-Invariant Learning of Deep Neural Networks”. ICML, 2022
> > >
> > > [5] Foret et al. “Sharpness-Aware Minimization for Efficiently Improving Generalization”. ICLR, 2021
> > >
> > > **3. "Just using Mixup and Randaugment seems not to be a strong baseline. What about stronger baseline such as the commonly-used DeiT ? Actually they mentioned that dropout hurts performance in their settings."**
> > >
> > > We use MixUp and RandAugment following the training recipe from [6] and we believe it is not a weak baseline since it contains half the augmentation techniques used by DeiT (DeiT uses MixUp, RandAugment, CutMix and random erase). In fact, the left panel in Figure 4 of [6] shows that with only MixUp and RandAugment, ViT-B16 can reach up to 83% test accuracy on ImageNet, which is similar to the 83.1% test accuracy achieved by DeiT-B (Table 7 of [7]).
> > >
> > > Due to the differences between DAMP and Dropout stated above, the fact that Dropout hurts performance of DeiT does not imply that DAMP would as well.
> > >
> > > [6] Steiner et al. "How to train your ViT? Data, Augmentation, and Regularization in Vision Transformers". TMLR 2022.
> > >
> > > [7] Touvron et al. "Training data-efficient image transformers & distillation through attention". ICML 2021.

---

> > > > ### Comment · Reviewer_XEeL · 2024-08-13
> > > >
> > > > Thank you to the authors for their response. Could you please elaborate further on the section regarding the same/different input nodes mentioned in point 1? It might be helpful to illustrate this with a specific mathematical formula.

---

> > > > > ### Author Response · Authors · 2024-08-13
> > > > >
> > > > > Thank you for engaging in the discussion on our work.
> > > > >
> > > > > We will demonstrate the differences between DAMP and Dropout/Variational Dropout (VD) below using a simple linear layer:
> > > > > * Original calculation of the forward pass without DAMP, Dropout or VD: $$\mathbf{y} = \mathbf{W}\cdot\mathbf{x}$$ where $\mathbf{y} \in \mathbb{R}^{m}, \mathbf{x} \in \mathbb{R}^{n}$ and $\mathbf{W} \in \mathbb{R}^{m \times n}$ and $\cdot$ is the matrix-vector product.
> > > > > * DAMP: Calculating the forward pass of the linear layer above with DAMP is as follows: $$\mathbf{y}\_{DAMP} = (\mathbf{W} \circ \boldsymbol{\xi})\cdot\mathbf{x}=\underbrace{\begin{bmatrix}W\_{1,1}\xi\_{1,1} & W\_{1,2}\xi\_{1,2} & \cdots & W\_{1,n}\xi\_{1,n} \\\ W\_{2,1}\xi\_{2,1} & W\_{2,2}\xi\_{2,2} & \cdots & W\_{2,n}\xi\_{2,n} \\\ \vdots & \vdots & \ddots & \vdots \\\ W\_{m,1}\xi\_{m,1} & W\_{m,2}\xi\_{m,2} & \cdots & W\_{m,n}\xi\_{m,n}\end{bmatrix}}\_{\mathbf{W}\circ\boldsymbol{\xi}}\cdot\mathbf{x}$$ where $\mathbf{W},\boldsymbol{\xi} \in \mathbb{R}^{m\times n}$ are two matrices of the same dimension and $\circ$ denotes the Hadamard product. Here each element $\xi\_{i,j}$ of $\boldsymbol{\xi}$ is a **different** Gaussian random variable with distribution $\mathcal{N}(1, \sigma^2\_{i,j})$. In DAMP, for simplicity, we set $\sigma\_{i,j}^2 = \sigma^2$ for all $i\in[1,m]$ and $j\in[1,n]$. Therefore, DAMP **multiplies each individual weight with its own Gaussian random variable**.
> > > > > * Dropout/VD: Calculating the forward pass of the linear layer above with Dropout/VD is as follows: $$\mathbf{y}\_{Dropout/VD}=\mathbf{W}\cdot(\mathbf{x}\circ\mathbf{r})=\mathbf{W}\cdot\begin{bmatrix}x\_1r\_1 \\\ x\_2r\_2 \\\ \vdots \\\ x\_nr\_n\end{bmatrix}=\begin{bmatrix}W\_{1,1}r\_1 & W\_{1,2}r\_2 & \cdots & W\_{1,n}r\_n \\\ W\_{2,1}r\_1 & W\_{2,2}r\_2 & \cdots & W\_{2,n}r\_n \\\ \vdots & \vdots & \ddots & \vdots \\\ W\_{m,1}r\_1 & W\_{m,2}r\_2 & \cdots & W\_{m,n}r\_n\end{bmatrix}\cdot\mathbf{x}$$ where $\mathbf{r}\in\mathbb{R}^n$ has the same dimension as input $\mathbf{x}$. For Dropout, each element $r\_i$ of $\mathbf{r}$ is a random variable following a Bernoulli distribution, while for VD, they follow the Gaussian distribution. As the equation has demonstrated, Dropout and VD **share the same multiplicative random variable for all weights connecting to the same input node**, i.e., all the weights in each column $i$ of matrix $\mathbf{W}$ share the same random variable $r\_i$.
> > > > >
> > > > > We hope that these equations are helpful to further understand the differences between DAMP and Dropout/VD, and we are happy to answer any follow-up questions that you might have.

---

> ### Comment · Reviewer_XEeL · 2024-08-14
>
> I appreciate the authors for their prompt response. It appears that DMAP is more similar to variational DropConnect [1], as it involves multiplying random variables with weights rather than activations. This approach also seems akin to variational inference in Bayesian Neural Networks (BNNs), where independent Gaussian variables are multiplied or added to weights in each forward process in practice, as seen in recent methods like SSVI [2].
>
> I would be inclined to support acceptance if the authors could clarify these 2 points further in future revisions. This work seems to provide valuable guidance and aims to scale these techniques to larger models and datasets, with a specific benefit in robustness.
>
> 1. Rimella, Lorenzo, and Nick Whiteley. "Dynamic Bayesian Neural Networks." arXiv preprint arXiv:2004.06963 (2020).
>
> 2. Li, Junbo, et al. "Training Bayesian Neural Networks with Sparse Subspace Variational Inference." arXiv preprint arXiv:2402.11025 (2024).

---

> > ### Author Response · Authors · 2024-08-14
> >
> > Thank you for your response.
> >
> > **1. On the connection between DAMP and variational inference methods such as variational DropConnect [1] and SSVI [2]**
> >
> > DAMP indeed could be interpreted as performing variational inference. This can be seen from the objective function of DAMP (Eq. 16 in our paper), which we restate below:
> > $$\mathcal{L}\_{DAMP}(\boldsymbol{\omega};\mathcal{S})=\underbrace{\mathbb{E}\_{\boldsymbol{\xi}\sim p(\boldsymbol{\xi})}\left[\mathcal{L}(\boldsymbol{\omega}\circ\boldsymbol{\xi};\mathcal{S})\right]}\_{\text{expected loss}} + \underbrace{\frac{\lambda}{2}||\boldsymbol{\omega}||\_F^2}\_{\text{$L\_2$-regularization}}$$
> > where:
> > * $\boldsymbol{\omega}$ is the model's weights.
> > * $\boldsymbol{\xi}$ is the vector of the multiplicative random variables with the same dimension as $\boldsymbol{\omega}$ whose distribution is $p(\boldsymbol{\xi})=\mathcal{N}(\boldsymbol{1},\sigma^2\mathbf{I})$.
> > * $\circ$ denotes the Hadamard product.
> > * $\mathcal{S}$ is the training data.
> > * $||\cdot||\_F^2$ denotes the Frobenius norm.
> > * $\mathcal{L}$ is **the original loss function which is the negative log-likelihood loss** for classification tasks.
> >
> > Therefore, the **expected loss** in the above equation can be interpreted as the expected log-likelihood term of the evidence lower-bound (ELBO) (the loss function used in variational inference) where the variational distribution is the distribution of the multiplicative random variables $p(\boldsymbol{\xi})$. In fact, since we don't optimize the variational distribution $p(\boldsymbol{\xi})$ and keep it fixed at $\mathcal{N}(\boldsymbol{1},\sigma^2\mathbf{I})$ throughout training, **minimizing the loss function $\mathcal{L}\_{DAMP}$ is equivalent to minimizing the negative ELBO plus an $L\_2$-regularization term** (since the KL divergence typically presented in the ELBO has no effect on training when we don't optimize the variational distribution.)
> >
> > From this perspective, DAMP is indeed related to [1] and [2] as you have suggested. Furthermore, our work presents directions to scale these variational approaches to large models and datasets, as you have concluded above. We will include the main points of this  discussion in the revised version of our paper.
> >
> > We hope that this response is helpful to further understand our method.
> >
> > [1] Rimella, Lorenzo, and Nick Whiteley. "Dynamic Bayesian Neural Networks." arXiv preprint arXiv:2004.06963 (2020).
> >
> > [2] Li, Junbo, et al. "Training Bayesian Neural Networks with Sparse Subspace Variational Inference." arXiv preprint arXiv:2402.11025 (2024).

---

### Author Rebuttal · Authors · 2024-08-06

We want to thank the reviewers for their time and for providing us with thoughtful comments which help us improve our work. In this global response, we first provide a brief summary of our paper. We then present additional experiment results, which are included in the attached PDF file.
## Paper summary:

The aim of our paper is to introduce a novel, efficient training method that enhances neural network (NN) robustness against input perturbations without incurring additional costs. We begin by presenting theoretical analyses demonstrating that input space corruptions can be simulated using multiplicative weight perturbations (MWPs) in the weight space. We argue that training under these simulated corruptions would allow models to become more robust. From these insights, we propose Data Augmentation via Multiplicative Perturbations (DAMP), which optimizes NNs under random Gaussian MWPs. At each training iteration, DAMP multiplies **each individual weight with its own random variable $\mathcal{N}(1, \sigma^2)$**. Additionally, we offer a new perspective on ASAM [1], revealing that it also produces robust models by optimizing under adversarial MWPs. Notably, while DAMP maintains the same training time as standard optimizers like Adam, ASAM *doubles* this duration. Our experimental results, conducted on both small datasets (CIFAR, TinyImageNet) and large-scale scenarios (ResNet50 and ViT-S16 on ImageNet), demonstrate that DAMP outperforms baselines in robustifying neural networks against a wide range of corruptions.
## New results:

The additional results in the PDF file demonstrate the following properties of DAMP:
* Domain-agnostic:  DAMP can be used in non-computer-vision domains as it can be used to finetune a large language model.
* Compatible with modern augmentations: DAMP can be combined with modern data augmentations to further enhance robustness.
* Capable of training large models: DAMP has no problem training ViT-B16 on ImageNet and achieves better results than AdamW.
* Same training costs as standard optimizers: We back up the claim that DAMP and standard optimizers like AdamW have the same training time by providing concrete numbers.
* Capable of improving robustness to various corruptions: We evaluate DAMP on additional types of corruptions including adversarial corruptions to verify that it is indeed effective on a wide range of corruptions.

We believe that these results will resolve all of the concerns that you had. Below, we outline the contents of the PDF file:
- Table A includes the results of using DAMP with LoRA [2] to finetune Mistral-7B [3] on the MedMCQA dataset [4]. These results show that DAMP consistently improves the test accuracy of the finetuned models, which answer concern of Reviewer **Vu1M** about the benefit of DAMP on non-computer-vision tasks, as well as concerns of Reviewer **XEeL** and **Vu1M** about the advantages of DAMP over domain-specific augmentations like MixUp [5] and RandAugment [6].
- Table B extends the results of ViT-S16 / ImageNet with basic Inception-style augmentations in our paper with additional evaluation results on ImageNet-D [7], ImageNet-A [8], ImageNet-Sketch [9], ImageNet-Cartoon [10], ImageNet-Drawing [10], and adversarial corruptions generated by FGSM [11]. These results show that DAMP produces the most robust model on average, which address the concerns of Reviewer **dM3x** and **dwU7** about robustness of DAMP to other types of input perturbations.
- Table B also includes the results averaged over all severity levels of ImageNet-C and ImageNet-$\bar{\mathrm{C}}$ and the runtime of each experiment as requested by Reviewer **Vu1M**. These runtimes show that ASAM and SAM indeed take roughly twice as long to train than DAMP and AdamW, and that DAMP and AdamW have the same runtime.
- Table C shows the behavior of DAMP in the ViT-S16 / ImageNet experiments with basic Inception-style augmentations under different standard deviations $\sigma$ of the Gaussian random multiplicative weight perturbations (MWPs) as requested by Reviewer **dM3x**. These results show that as $\sigma$ increases, the accuracy of DAMP improves up to its maximum value and then degrades afterwards.
- Table D presents results of DAMP and baselines when training ViT-S16 and ViT-B16 on ImageNet with MixUp [5] and RandAugment [6], demonstrating that: (i) DAMP can be combined with modern data augmentations (DAs) to further enhance robustness; (ii) DAMP is capable of training large models like ViT-B16; (iii) given the same amount of training time, it is better to train a large model (ViT-B16) using DAMP than to train a smaller model (ViT-S16) using SAM/ASAM. This addresses the concerns of Reviewer **dwU7** and **Vu1M** regarding the advantages of DAMP over SAM/ASAM, and concern of Reviewer **Vu1M** about the compatibility of DAMP with modern DA techniques.

[1] Kwon et al. "ASAM: Adaptive Sharpness-Aware Minimization for Scale-Invariant Learning of Deep Neural Networks". ICML 2022.

[2] Hu et al. "LoRA: Low-Rank Adaptation of Large Language Models". ICLR 2022.

[3] Jiang et al. "Mistral-7B". arXiv preprint arXiv:2310.06825, 2023.

[4] Pal et al. "MedMCQA : A Large-scale Multi-Subject Multi-Choice Dataset for Medical domain Question Answering". CHIL 2022.

[5] Zhang et al. "mixup: Beyond Empirical Risk Minimization". ICLR 2018.

[6] Cubuk et al. "RandAugment: Practical automated data augmentation with a reduced search space". NeurIPS 2020.

[7] Zhang et al. "ImageNet-D: Benchmarking Neural Network Robustness on Diffusion Synthetic Object". CVPR 2024.

[8] Hendrycks et al. "Natural Adversarial Examples". CVPR 2021.

[9] Wang et al. "Learning Robust Global Representations by Penalizing Local Predictive Power". NeurIPS 2019.

[10] Salvador et al. "ImageNet-Cartoon and ImageNet-Drawing: two domain shift datasets for ImageNet". ICML 2022 Shift Happens Workshop.

[11] Goodfellow et al. "Explaining and Harnessing Adversarial Examples". arXiv preprint arXiv:1412.6572, 2014.

---

### Decision · Program_Chairs · 2024-09-25

**Decision:**

Accept (spotlight)

**Comment:**

The paper tackles the problem of "robustness" of deep neural networks (DNN) in the presence of "corrupted" images. The main intuition/contribution is a method that improves the robustness of DNN to corrupted images by manipulating the weights of the DNN during its training phase---thereby suggesting that there is some sort of correlation between perturbations in the "input space" (i.e., in the images analysed by the DNN) and those in the "weight space" (i.e., in the weights of the DNN). Experiments on various benchmark datasets, as well as various theoretical proofs, support the effectiveness of the proposed method. Moreover, **the rebuttal phase also allowed to shed light on the fact that the proposed method is not specific to DNN analysing images** (as demonstrated in an experiment on the MedMCQA dataset), thereby further showcasing the generality and "improvement" of the proposed solution over existing alternatives.


Four reviews were submitted for this paper. Two reviewers (with confidence 3 and 4) originally recommended a "5-Borderline Accept" and then increased to a "6-Weak Accept" after interacting with the authors. Another reviewer (with a confidence of 2) proposed a "6-Weak Accept" and maintained their score after the rebuttal phase. A fourth reviewer recommended a "4-Borderline Reject" (with a high-confidence of 4), rating the paper with 1-2-1 for Soundness, Presentation and Contribution, but the exchanges with the authors led such a reviewer to improve their Soundness and Contribution ratings (to a 3 and to a 2) and also to recommend a 5 "Borderline Accept". Hence, it can be said that the author-reviewer interactive phase played a pivotal role in clarifying lingering doubts as well as in increasing the quality of this submission. This is a positive sign of a constructive peer-review.

No factual errors were found by the reviewers. The code is also made public. Moreover, all reviewers agreed in that the paper provides a meaningful contribution to the state of the art. Some praised the extensive experiments, or the theoretical foundations, or even the presentation of the paper---all of which being aspects that I also believe to be significant strengths of this submission.

Due to the above, I recommend acceptance of this work---with a "spotlight". I could not identify glaring weaknesses, and the combination of sound experiments, theoretical analyses, good presentation, nice intuition, and generalizability make me believe that this paper can spearhead many discussions in the field.